https://doi.org/10.1038/s41467-019-10518-0　　**OPEN**

# Phytochrome activates the plastid-encoded RNA polymerase for chloroplast biogenesis via nucleus-to-plastid signaling

Chan Yul Yoo [ID] [1], Elise K. Pasoreck[1], He Wang[1], Jun Cao[2], Gregor M. Blaha[3], Detlef Weigel[2] & Meng Chen [ID] [1]

Light initiates chloroplast biogenesis by activating photosynthesis-associated genes encoded by not only the nuclear but also the plastidial genome, but how photoreceptors control plastidial gene expression remains enigmatic. Here we show that the photoactivation of phytochromes triggers the expression of photosynthesis-associated plastid-encoded genes (*PhAPG*s) by stimulating the assembly of the bacterial-type plastidial RNA polymerase (PEP) into a 1000-kDa complex. Using forward genetic approaches, we identified REGULATOR OF CHLOROPLAST BIOGENESIS (RCB) as a dual-targeted nuclear/plastidial phytochrome signaling component required for PEP assembly. Surprisingly, RCB controls *PhAPG* expression primarily from the nucleus by interacting with phytochromes and promoting their localization to photobodies for the degradation of the transcriptional regulators PIF1 and PIF3. RCB-dependent PIF degradation in the nucleus signals the plastids for PEP assembly and *PhAPG* expression. Thus, our findings reveal the framework of a nucleus-to-plastid anterograde signaling pathway by which phytochrome signaling in the nucleus controls plastidial transcription.

[1] Department of Botany and Plant Sciences, Institute for Integrative Genome Biology, University of California, Riverside, CA 92521, USA. [2] Department of Molecular Biology, Max Planck Institute for Developmental Biology, 72076 Tübingen, Germany. [3] Department of Biochemistry, University of California, Riverside, CA 92521, USA. Correspondence and requests for materials should be addressed to M.C. (email: meng.chen@ucr.edu)

Extranuclear genomes in organelles such as mitochondria and plastids define the eukaryotic cell. While perturbing the activity of the mitochondrial genome leads to human pathologies, altering plastidial gene expression can kill plants[1–3]. The plastidial genome carries 100–120 genes encoding essential components of the plastidial transcriptional, translational, and photosynthetic apparatuses[4], and therefore, the regulation of plastidial gene expression is critical for the biogenesis of photosynthetically active chloroplasts[5,6]. However, because only <10% of plastid proteins are encoded by the plastidial genome, and the rest by the nuclear genome, the plastid is genetically semi-autonomous[5,6]. Plastidial gene expression is thought to be determined by the developmental program of the host cell[5], but the cell signaling mechanisms that control plastidial gene expression remains elusive.

The greening, or chloroplast biogenesis, of flowering plants (angiosperms) occurs only in the presence of light[7]. Light promotes chloroplast biogenesis by controlling two distinct biological processes in the plant cell. Light turns on chlorophyll biosynthesis directly in plastids by activating protochlorophyllide oxidoreductase to catalyze the final step of chlorophyll biosynthesis, the conversion of protochlorophyllide to chlorophyll $a$[8]. More importantly, as an environmental signal, light reprograms hundreds of genes in the nucleus to initiate the developmental transition from heterotrophic growth supported by seed-stored energy to autotrophic growth, which relies on photosynthesis[7,9]. In dicotyledonous plants, such as *Arabidopsis thaliana*, young seedlings that germinate underground adopt a dark-grown developmental program called skotomorphogenesis or etiolation, which inhibits leaf development and promotes elongation of the embryonic stem (hypocotyl), allowing seedlings to emerge quickly from the soil into the light. Plastids in the leaves of dark-grown seedlings differentiate into photosynthetically inactive, non-green etioplasts. Upon emerging from the soil, seedlings transition to a light-grown developmental program called photomorphogenesis, which restricts hypocotyl elongation, and promotes leaf development and chloroplast biogenesis.

Light establishes photomorphogenesis through the massive transcriptional reprogramming of the nuclear genome. Light is first perceived by a suite of photoreceptors, including the red (R)- and far-red (FR)-light-sensing phytochromes (PHYs), which play an essential role in chloroplast biogenesis[10–12]. In *Arabidopsis*, PHYs are encoded by five genes, *PHYA-E*, among which the gene products of *PHYA* and *PHYB* are the predominant sensors of continuous FR and R light, respectively[13–15]. PHYs utilize a covalently attached linear tetrapyrrole as a chromophore to sense light through conformational switches between the R-light-absorbing inactive Pr form and the FR-light-absorbing active Pfr form[16]. PHYs are synthesized in the Pr form in the cytoplasm. Upon photoactivation to the Pfr form, PHYs accumulate in the nucleus and localize to punctate subnuclear foci named photobodies[17–19]. The size and number of photobodies are directly regulated by light quality and quantity[20,21]. Under strong light, PHYB-GFP is confined to only a few large photobodies of 0.7–2 μm in diameter[20,21]. Shifting the equilibrium of PHYs toward the inactive Pr form under low light or shade conditions induces PHYB-GFP to localize to tens of smaller photobodies of 0.1–0.7 μm in diameter[20,21]. PHYs colocalize on photobodies with a group of phytochrome-interacting transcription factors, the PIFs[22,23]. The PIF family of transcriptional regulators include eight members, PIF1, PIF3-8, and PIL1 (PIF3-Like1); they are repressors of photomorphogenesis[24–26]. Most PIFs accumulate to high levels in dark-grown seedlings, where they promote hypocotyl elongation by activating growth-relevant genes and inhibit chloroplast biogenesis by repressing photosynthesis-associated nuclear-encoded genes (*PhANGs*)[24,25]. Photoactivated PHYs interact directly with PIFs and trigger their degradation, which is a central mechanism in reprogramming the nuclear genome to initiate photomorphogenesis[24,25,27]. Although the cellular mechanism of PIF degradation is still not fully understood, accumulating evidence indicates that the formation of large photobodies is closely associated with PIF3 degradation[21,28,29].

Chloroplast biogenesis also requires the activation of photosynthesis-associated plastid-encoded genes (*PhAPGs*), which encode essential components of the photosynthetic apparatus, such as the large subunit of the carbon fixation enzyme ribulose-1,5-bisphosphate carboxylase/oxygenase (rbcL) and the photosystem II reaction center D1 protein (psbA)[4]. Plastidial genes are transcribed by two types of RNA polymerases: a phage-type nuclear-encoded RNA polymerase (NEP) and a bacterial-type plastid-encoded RNA polymerase (PEP)[30,31]. While the NEP preferentially transcribes housekeeping genes, including plastid ribosomal RNAs and the core subunits of the PEP, the PEP transcribes *PhAPGs*[32]. *PhAPGs* are induced transcriptionally by light and PHYs[33,34]. Because PHYs do not localize to the plastids, they need to control plastidial transcription from outside of the plastids, but the signaling mechanism by which PHYs control plastidial transcription is largely unknown. Here we show that PHYs trigger light-dependent assembly of the PEP into a 1000-kDa protein complex for *PhAPG* transcription. Using a forward genetic screen, we identified REGULATOR OF CHLOROPLAST BIOGENESIS (RCB) as a necessary PHY signaling component that activates the assembly and activation of the PEP from the nucleus by promoting photobody biogenesis and PIF degradation. Intriguingly, PIF degradation in the nucleus signals the plastids to assemble and activate the PEP. Thus, this study reveals the framework of a nucleus-to-plastid light signaling mechanism linking nuclear PHY signaling and the control of the PEP for *PhAPG* transcription during chloroplast biogenesis.

## Results

**Phytochromes trigger light-dependent PEP assembly**. Chloroplast biogenesis in the light is principally controlled by PHYs. Knocking out all *PHY*s completely blocks greening in rice and dramatically impairs greening in *Arabidopsis* in R light (Fig. 1a)[10–12]. The total chlorophyll contents in R-light-grown *phyABCDE*, *phyA-211/phyB-9*, and *phyB-9* mutants were reduced by 96.4%, 63.7%, and 59.6%, respectively, compared with that in the wild-type (Fig. 1b). These results indicate that PHYs, particularly PHYA and PHYB, play critical roles in initiating chloroplast biogenesis. It is important to note that *phyB-9* carries a second-site mutation that partially contributes to its greening phenotype, but this mutation is not present in *phyA-211/phyB-9*[35]. Further supporting the important role of PHYA and PHYB in chloroplast biogenesis, when dark-grown seedlings were illuminated by R light during the dark-to-light transition, greening and chlorophyll accumulation in *phyA-211/phyB-9* were significantly attenuated (Fig. 1c, d). To investigate a possible connection between PHY signaling and the regulation of plastidial gene expression, we examined PEP- and NEP-dependent genes in Col-0 and *phy* mutants. The steady-state mRNA levels of three PEP-dependent *PhAPGs*, *psbA*, *psbB*, and *rbcL*, in Col-0 increased more than sixfold in the light than in darkness (Fig. 1e) and more than sevenfold within 48 h during the dark-to-R-light transition (Fig. 1f). In contrast, the expression of three NEP-dependent marker genes, *rpoB*, *rpoC1*, and *rpl20*, in Col-0 did not change substantially in continuous R light compared with that in darkness nor during the dark-to-R-light transition (Fig. 1e, f). *PhAPGs* also failed to be activated in *phyB-9*, *phyA-211/phyB-9*, and *phyABCDE* mutants grown in continuous R light as well as in *phyA-211/phyB-9* during the dark-to-R-light transition (Fig. 1e,

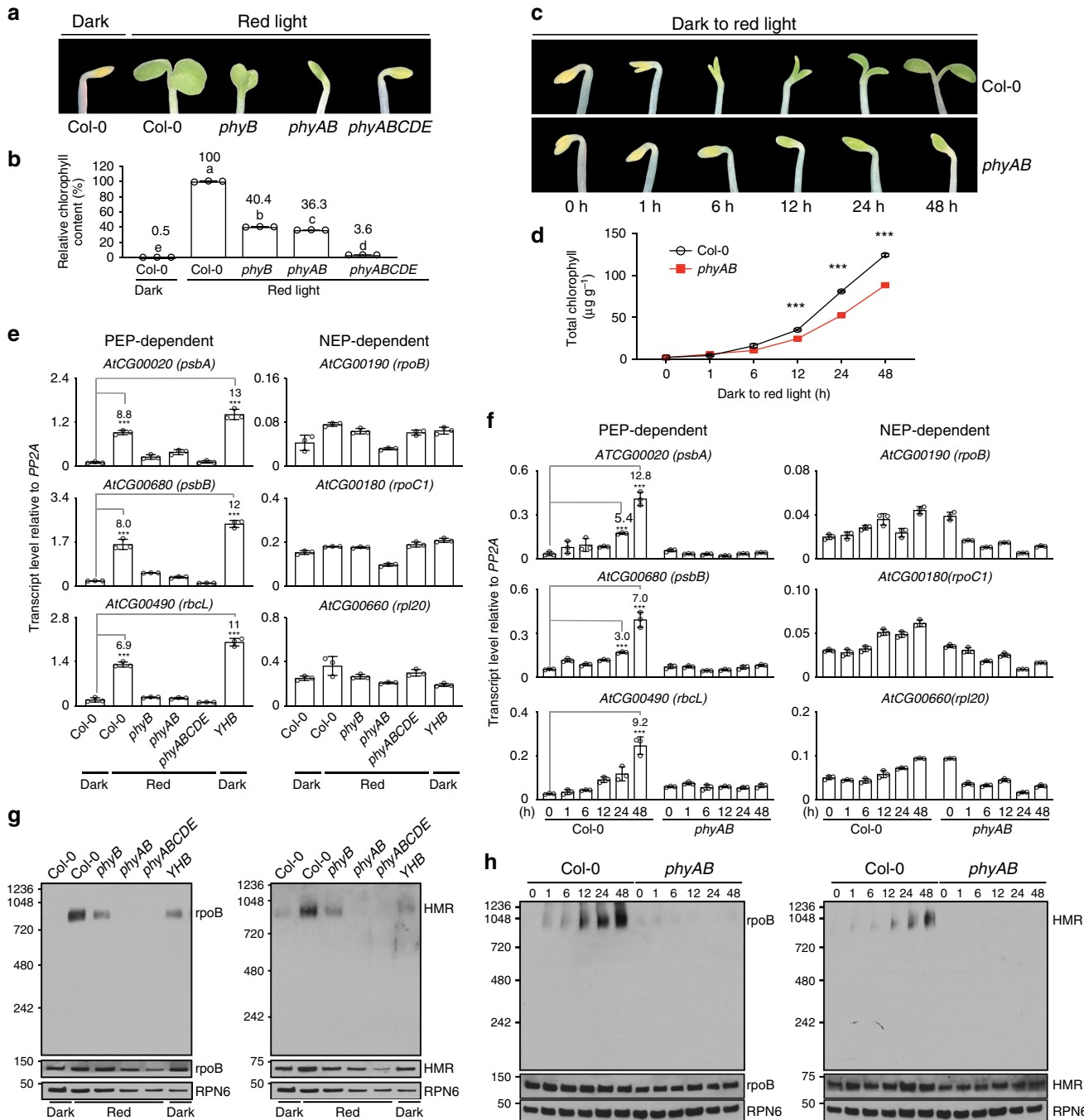

**Fig. 1** Phytochromes activate *PhAPG*s by promoting the assembly of the PEP. **a** Images of embryonic leaves from 4-d-old Col-0, *phyB-9* (*phyB*), *phyA-211/ phyB-9* (*phyAB*), and *phyA/phyB/phyC/phyD/phyE* (*phyABCDE*) seedlings grown in either darkness or 10 μmol m$^{-2}$ s$^{-1}$ R light. **b** Total chlorophyll levels in seedlings shown in (**a**). Different letters denote statistically significant differences in chlorophyll content (ANOVA, Tukey's HSD, $p \leq 0.001$). **c** Images of embryonic leaves of 4-d-old Col-0 and *phyAB* seedlings from the indicated time points after dark-grown seedlings were illuminated with 10 μmol m$^{-2}$ s$^{-1}$ R light. **d** Total chlorophyll levels in Col-0 and *phyAB* seedlings during the dark-to-light transition described in (**c**). *** Indicates a statistically significant difference between Col-0 and *phyAB* (Student's *t*-test, $p \leq 0.001$). **e**, **f** qRT-PCR results showing the transcript levels of representative PEP- and NEP-dependent genes in 4-d-old indicated lines grown in either darkness or 10 μmol m$^{-2}$ s$^{-1}$ R light (**e**) or during the dark-to-light transition described in (**c**) (**f**). The transcript levels were calculated relative to those of *PP2A*. Fold changes are shown only for the samples exhibiting greater than twofold changes compared with the values of dark-grown Col-0 (***, Student's *t*-test, $p \leq 0.001$). **g**, **h** Immunoblots showing the levels of the PEP complex as well as rpoB and HMR proteins in 4-d-old indicated lines (**g**) or during the dark-to-light transition (**h**). Total protein was isolated under either native or denaturing conditions and resolved by blue-native PAGE or SDS-PAGE, respectively. PEP complex on blue-native gels and denatured rpoB and HMR separated by SDS-PAGE were detected by immunoblots using antibodies against rpoB and HMR. RPN6 was used as a loading control. For (**b**), (**c**), (**e**), and (**f**), error bars represent SD of three biological replicates. The source data underlying the chlorophyll measurements in (**b**) and (**d**), the qRT-PCR analysis in (**e**) and (**f**), and immunoblots in (**g**) and (**h**) are provided in the Source Data file

f), indicating that PHYs are required for *PhAPG*s activation. Consistent with these results, in a constitutively active *phyB* mutant, *YHB*, which carries a Y276H mutation in PHYB's photosensory chromophore attachment domain that locks PHYB in an active form[36], *PhAPG*s became active in the dark (Fig. 1e), indicating that activation of PHYB alone is sufficient to induce *PhAPG* expression. In contrast to the PEP-dependent genes, NEP-dependent genes were not altered by more than twofold in the *phy* mutants (Fig. 1e, f). Together, these results provide evidence that PHYs can trigger the plastid to activate the expression of *PhAPG*s by the PEP. These results prompted us to investigate how PHY signaling activates the PEP.

The PEP comprises prokaryotic α, β, β′, β″ core subunits and a sigma factor surrounded by twelve plant-specific PEP-associated proteins that are essential for its activity[1,37,38]. Extensive biochemical studies have demonstrated that a large fraction of the PEP is tightly associated with DNA and forms multisubunit complexes[37–40]. We therefore tested whether the formation of the *Arabidopsis* PEP complex is influenced by light and PHY signaling. To that end, we resolved the PEP complex from *Arabidopsis* by blue-native-gel electrophoresis and monitored its size by immunoblotting using antibodies against either the core β subunit, rpoB, or a PEP-associated protein, HEMERA (HMR)/pTAC12. We found that the PEP in R light-grown Col-0 forms a 1000-kDa complex, which could be detected by anti-rpoB and anti-HMR antibodies (Fig. 1g). Strikingly, although rpoB and HMR were present in Col-0 seedlings grown in both light and dark conditions, the 1000-kDa PEP complex was absent in the dark (Fig. 1g). The core subunits of the bacterial RNA polymerase is self-sufficient to form a functional complex[41]. Intriguingly, we did not observe a smaller complex of the bacterial-like core subunits of the PEP in either the dark or the light, suggesting that the core subunits of the *Arabidopsis* PEP require additional factors, likely the plant-specific PEP-associated proteins, for its assembly. During the dark-to-R-light transition, the PEP complex appeared within 1 h after light exposure and increased to a steady-state level in 48 h (Fig. 1h). These results indicate that PEP assembly is triggered by light and correlates with *PhAPG* expression (Fig. 1e–h). The amount of PEP complex was greatly reduced in *phyB-9* seedlings and was undetectable in *phyA-211/phyB-9* and *phyABCDE* mutants in the light (Fig. 1g). PEP assembly was blocked in *phyA-211/phyB-9* during the dark-to-R-light transition (Fig. 1h). Also, PEP assembly was activated in the *YHB* mutant in the dark (Fig. 1g). Together, these results demonstrate that the photoactivation of PHYs, particularly PHYA and PHYB, causes PEP components to assemble into a 1000-kDa complex, providing a mechanism for activating the PEP by light.

**Identification of RCB**. A major challenge that has hindered the discovery of the mechanism of PHY signaling in controlling chloroplast biogenesis is the lack of an efficient forward genetic screening strategy that can distinguish chloroplast-deficient signaling mutants from other albino mutants of genes encoding essential components of the chloroplast[5]. Our recent genetic studies of early PHY signaling have found HMR, which is so far the only identified PHY signaling component essential for chloroplast biogenesis[28]. The *hmr* mutant represents the founding member of a mutant class with a combination of long-hypocotyl and albino seedling phenotypes, which are indicative of deficiencies in PHY signaling and chloroplast biogenesis, respectively[28,42]. Albino mutants had been previously considered uninteresting in the context of light signaling because historically, chlorophyll-deficient mutants had been shown to retain normal PHY-mediated hypocotyl responses[43,44]. As a result, the entire

class of tall-and-albino mutants like *hmr* had been ignored[42]. We hypothesized that some of the tall-and-albino mutants could define missing PHY signaling components in the signaling pathway that activates *PhAPG* transcription.

We therefore performed a forward genetic screen for *hmr-like* mutants with tall-and-albino phenotypes in monochromatic R light. The screen was conducted using *PBG* (PHYB-GFP), a transgenic line in the null *phyB-5* background complemented with functional PHYB-GFP[17]. This design allowed us to easily assess whether the diagnostic signaling event of photobody formation is impaired in the mutants. From 2,000 N-ethyl-N-nitrosourea or ethyl methanesulfonate mutagenized F$_2$ *PBG* families, we identified 23 tall-and-albino mutants. In this study, we have focused on two mutations in the same complementation group. We named this locus *Regulator of Chloroplast Biogenesis* (*RCB*). Both *rcb-1/PBG* and *rcb-2/PBG* seedlings had elongated hypocotyls and albino embryonic leaves in R light (Fig. 2a, b), suggesting that RCB is required for PHY-mediated photoinhibition of hypocotyl elongation and chloroplast biogenesis.

We mapped the mutations co-segregating with the tall-and-albino phenotype of *rcb-1/PBG* and *rcb-2/PBG* using SHOREmap to the same gene At4g28590. *rcb-1/PBG* features a 1-bp deletion in chromosome 4 at position 14,126,279 that generates a frameshift in the second exon of At4g28590. *rcb-2/PBG* contains a G-to-A substitution at nucleotide 14,216,245 of chromosome 4, which introduces a premature stop codon at codon 195 in At4g28590 (Fig. 2c). We generated an antibody against the gene product of At4g28590 and found that neither *rcb-1/PBG* nor *rcb-2/PBG* accumulated the protein product of At4g28590 (Supplementary Fig. 1a), indicating that they are null alleles. Expressing the cDNA of At4g28590 rescued the tall-and-albino phenotype of *rcb-1/PBG* (Supplementary Fig. 1b, c). In addition, we obtained a T-DNA insertion allele *rcb-10*, SALK_075057[45], which carries a T-DNA insertion in the second intron of At4g28590 after nucleotide 14,125,739 (Fig. 2c). The gene product of At4g28590 did not accumulate in *rcb-10* (Supplementary Fig. 1a), indicating that it is a null allele. Similar to *rcb-1/PBG* and *rcb-2/PBG*, *rcb-10* was also tall and albino (Fig. 2d, e). Together, these results demonstrate that At4g28590 is *RCB*.

*RCB* encodes a 331-amino-acid protein with three recognizable motifs (Fig. 2c): a transit peptide predicted by ChloroP1.1[46], a monopartite nuclear localization signal (NLS) predicted by NLS mapper[47], and a carboxy terminal thioredoxin-like fold recognized by Phyre2[48]. RCB has been identified previously because of its essential role in chloroplast biogenesis, and it was shown that *PhAPG* expression was impaired in the *rcb* mutant[49–52]. However, the precise function of RCB is still unknown. The expression of *PhAPG*s was downregulated in *rcb-1/PBG*, *rcb-2/PBG*, and *rcb-10*, whereas NEP-dependent genes were upregulated (Fig. 2f)—a characteristic of mutants defective specifically in the PEP[37]. The altered expression of PEP- and NEP-dependent genes in *rcb-1/PBG* was also rescued by expressing RCB (Supplementary Fig. 1d). Interestingly, we found that all three *rcb* mutants failed to stimulate the assembly of the 1000-kDa PEP complex in the light (Fig. 2g). These results indicate that RCB mediates *PhAPG* activation by promoting PEP assembly.

**RCB is required for phytochrome signaling**. A role of RCB in PHY signaling has never been revealed previously. We therefore wanted to determine if and how RCB participates in PHY signaling. To that end, we analyzed the hypocotyl elongation responses of the *rcb* mutants in continuous FR and R light to assess their effectiveness in PHYA and PHYB signaling, respectively[53]. These experiments showed that *rcb-10* and *rcb-1/PBG* were hyposensitive to FR and R light (Fig. 3a–d). The *rcb* mutant

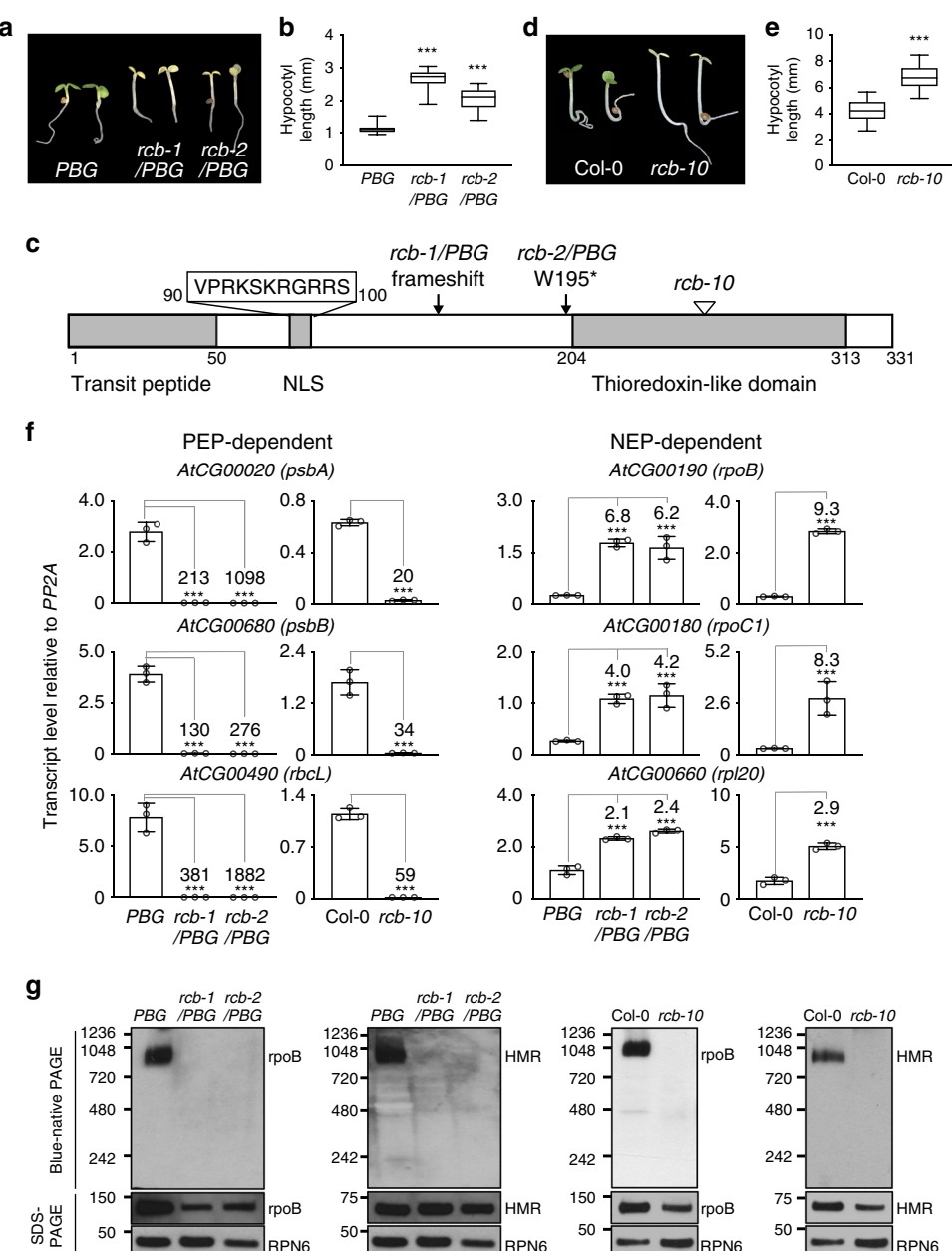

**Fig. 2** Identification of RCB as a PHY signaling component required for PEP assembly. **a** Images of 4-d-old *PBG*, *rcb-1/PBG*, and *rcb-2/PBG* seedlings grown in 10 μmol m$^{-2}$ s$^{-1}$ continuous R light. **b** Box-and-whisker plots showing hypocotyl measurements of seedlings in (**a**). The boxes represent from the 25th to the 75th percentiles, and the bars equal the median values. *** Indicates a statistically significant difference from the value of *PBG* (Student's *t*-test, *p* ≤ 0.001). **c** Schematic illustration of the predicted domain structure of RCB. The mutations of the *rcb* alleles are indicated. NLS, nuclear localization signal. **d** Representative images of 4-d-old Col-0 and *rcb-10* seedlings grown in 10 μmol m$^{-2}$ s$^{-1}$ R light. **e** Box-and-whisker plots showing hypocotyl measurements of seedlings in (**d**). The boxes represent from the 25th to the 75th percentiles, and the bars equal the median values. *** Indicates statistically significant differences from the value of Col-0 (Student's *t*-test, *p* ≤ 0.001). **f** qRT-PCR analyses of the transcript levels of representative PEP- and NEP-dependent genes in 4-d-old *PBG*, *rcb-1/PBG*, *rcb-2/PBG*, Col-0, and *rcb-10* seedlings grown in 10 μmol m$^{-2}$ s$^{-1}$ R light. Fold changes are shown for the *rcb* mutant samples exhibiting greater than twofold changes compared with the values of the corresponding parental-line samples (***, Student's *t*-test, *p* ≤ 0.001). Error bars represent SD of three biological replicates. **g** Immunoblots showing the levels of the PEP complex (blue-native PAGE) as well as total rpoB and HMR proteins (SDS-PAGE) in 4-d-old *PBG*, *rcb-1/PBG*, *rcb-2/PBG*, Col-0, and *rcb-10* seedlings grown in 10 μmol m$^{-2}$ s$^{-1}$ R light. RPN6 was used as a loading control. The source data underlying the hypocotyl measurements in (**b**) and (**e**), the qRT-PCR analysis in (**f**), and immunoblots in (**g**) are provided in the Source Data file

had a normal hypocotyl response in the dark (Supplementary Fig. 2a), and both *rcb-10/phyB-9* and *rcb-10/phyA-211* double mutants were not taller than *phyB-9* and *phyA-211*, respectively (Fig. 3e–h), indicating that the long-hypocotyl phenotype of the *rcb* mutants relies on the presence of PHY signaling. To further demonstrate RCB's role in PHY signaling, we crossed *rcb-1* to

the constitutively active *phyB* allele YHB[36]. The constitutive photomorphogenetic phenotype of YHB in the dark was partially suppressed in the *rcb-1/YHB* double mutant (Fig. 3i, j), confirming that RCB is required for PHYB signaling. To test whether RCB works in the same signaling pathway as HMR, we generated a *rcb-10/hmr-5* double mutant. The *rcb-10/hmr-5* double mutant

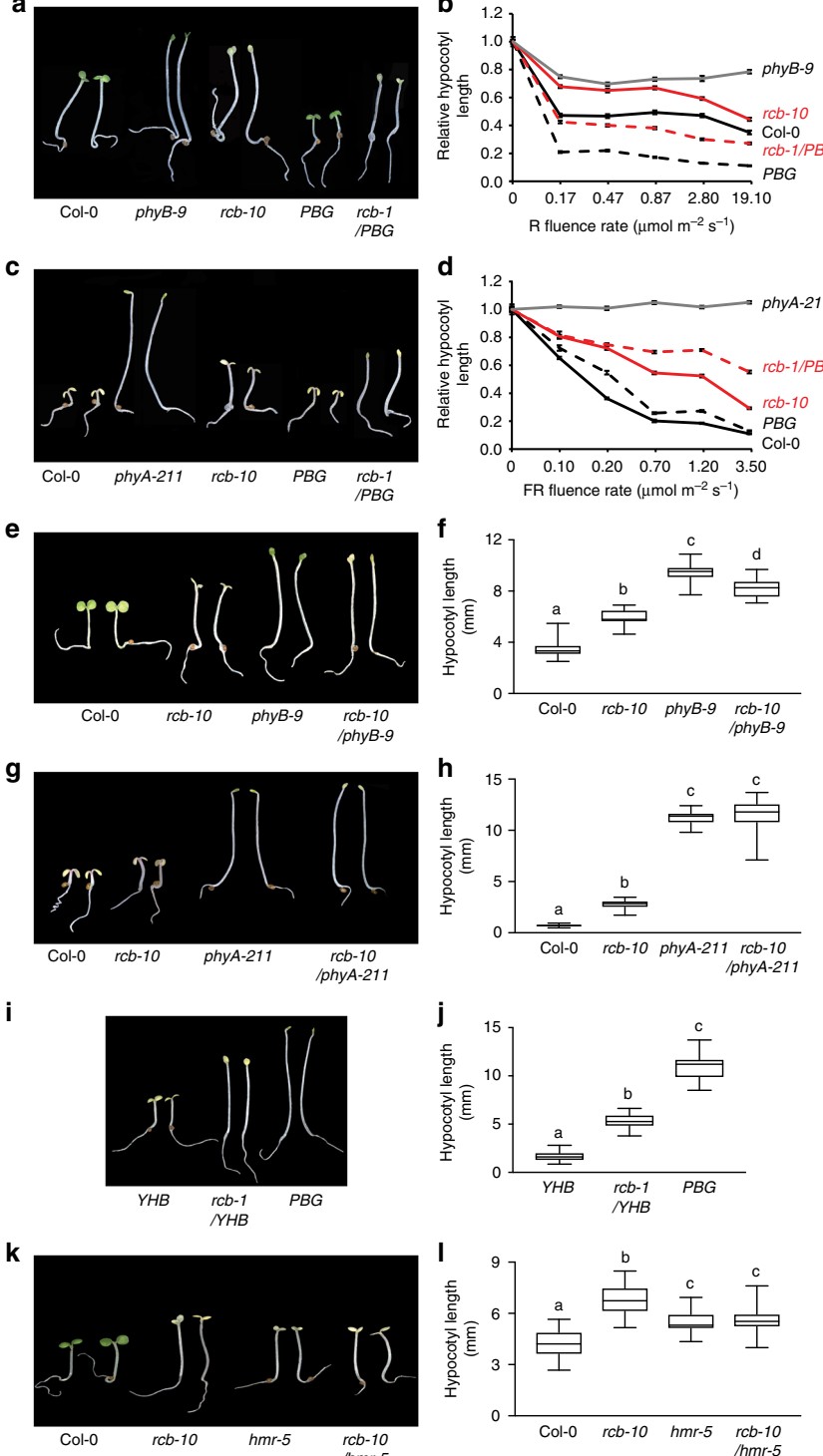

was not taller than either *rcb-10* or *hmr-5* (Fig. 3k, l), suggesting that RCB and HMR function in the same PHY-dependent pathway. Interestingly, *rcb-10/hmr-5* had the same hypocotyl length as *hmr-5* and was slightly shorter than *rcb-10* (Fig. 3k, l), which might suggest that for hypocotyl regulation, *hmr-5* is epistatic to *rcb-10*. In contrast to the clear defects in response to FR and R light, the *rcb* mutants had a normal hypocotyl response to blue light (Supplementary Fig. 2b), indicating that RCB is not required for cryptochrome-mediated blue light signaling. Together, these genetic results demonstrate that RCB participates in PHY-specific light signaling.

**RCB is a dual-targeted phytochrome signaling component.** RCB has been shown previously to localize exclusively to plastids[49,51,52]. How can we reconcile the requirement for RCB in PHYA and PHYB signaling in the nucleus with its protein localization to plastids? To further examine the function of RCB in PHY signaling, we asked whether RCB, with its putative NLS (Fig. 2c), is also targeted to the nucleus. We found that RCB-CFP was detectable in both the plastids and the nuclei of transiently transformed tobacco cells (Fig. 4a). In agreement with this result, a functional HA-His-tagged RCB expressed in *rcb-10* (RCB-HA-His/rcb-10) as well as the endogenous RCB were detected in both

**Fig. 3** RCB mediates PHY signaling. **a** Representative images of 4-d-old Col-0, *phyB-9*, *rcb-10*, *PBG*, and *rcb-1/PBG* seedlings grown in 0.5 μmol m$^{-2}$ s$^{-1}$ R light. **b** R light fluence response curves for Col-0, *phyB-9*, *rcb-10*, *PBG*, and *rcb-1/PBG* measured by growing seedlings for 4 days in darkness or a series of intensities of R light. Hypocotyl length in R light was calculated relative to the value of the corresponding lines in the dark. Error bars represent the SD of three biological replicates. **c** Representative images of 4-d-old Col-0, *phyA-211*, *rcb-10*, *PBG*, and *rcb-1/PBG* seedlings grown in 1 μmol m$^{-2}$ s$^{-1}$ FR light. **d** FR light fluence response curves for Col-0, *phyA-211*, *rcb-10*, *PBG*, and *rcb-1/PBG* measured by growing seedlings for 4 days in darkness or a series of intensities of FR light. Hypocotyl length in FR light was calculated relative to the value of the corresponding lines in the dark. Error bars represent SD of three biological replicates. **e** Representative images of 4-d-old Col-0, *rcb-10*, *phyB-9*, and *rcb-10/phyB-9* seedlings grown in 10 μmol m$^{-2}$ s$^{-1}$ R light. **f** Box-and-whisker plots showing hypocotyl measurements of seedlings in (**e**). **g** Representative images of 4-d-old Col-0, *rcb-10*, *phyA-211*, and *rcb-10/phyA-211* seedlings grown in 10 μmol m$^{-2}$ s$^{-1}$ FR light. **h** Box-and-whisker plots showing hypocotyl measurements of seedlings in (**g**). **i** Representative images of 4-d-old *YHB*, *rcb-1/YHB*, and *PBG* seedlings grown in darkness. **j** Box-and-whisker plots showing hypocotyl measurements of seedlings in (**i**). **k** Representative images of 4-d-old Col-0, *rcb-10*, *hmr-5*, and *rcb-10/hmr-5*, and *PBG* seedlings grown in 10 μmol m$^{-2}$ s$^{-1}$ R light. **l** Box-and-whisker plots showing hypocotyl measurements of seedlings in (**k**). For the box-and-whisker plots in (**f**), (**h**), (**j**), and (**l**), the boxes represent from the 25th to the 75th percentiles, and the bars equal the median values. Different letters denote statistically significant differences in hypocotyl length (ANOVA, Tukey's HSD, $p \leq 0.001$). The source data underlying the hypocotyl measurements in (**b**), (**d**), (**f**), (**h**), (**j**), and (**l**) are provided in the Source Data file

the nuclear and plastidial fractions (Fig. 4b, c and Supplementary Fig. 3). Surprisingly, the total, nuclear, and plastidial fractions of RCB-HA-His and endogenous RCB had similar molecular masses (Fig. 4b, c), even though plastid-localized RCB is expected to be smaller than nuclear RCB due to the removal of its transit peptide during plastid protein import. To examine the size of endogenous RCB more closely, we ran side-by-side in vitro translated full-length RCB and a series of N-terminal truncated RCB fragments. The endogenous RCB was significantly smaller than the full-length RCB and similar to RCBΔ51, lacking the N-terminal 51 amino acids (Fig. 4c). These results suggest that the transit peptide of RCB is around 50 amino acids (Fig. 2c) and imply that nuclear RCB also lacks the transit peptide, and therefore, it is possible that RCB is imported into plastids first before translocating to the nucleus[54]. Together, we conclude that RCB is targeted to not only plastids but also the nucleus. This raises the possibility that RCB participates directly in nuclear PHY signaling.

We next tested whether RCB is required for the PHY-dependent degradation of the PIFs. We focused on two well-characterized light-labile PIFs that are involved in chloroplast biogenesis—PIF1 and PIF3[55,56]. Interestingly, similar to *hmr-5*, PIF1 and PIF3 accumulated or failed to be completely degraded in *rcb-1/PBG* and *rcb-10* in the light (Fig. 4d), indicating that RCB is required for PIF1 and PIF3 degradation in the nucleus.

We have previously shown that the degradation of PIF1 and PIF3 depends on the transcriptional activator HMR and is coupled to the activation of a subset of growth-relevant PIF target genes (Fig. 4d)[28,57]. To examine whether RCB is involved in HMR-dependent regulation of PIF target genes, we performed microarray analysis to determine genome-wide RCB-dependent genes. We identified 992 genes that were changed (per-gene variance $p < 0.05$, common variance twofold change) between 4-d-old R-light-grown *rcb-10* and Col-0 (Supplementary Data 1). Most RCB-dependent genes (871, or 88%) were also differentially expressed in *hmr-5* (Fig. 4e and Supplementary Data 2)[57]. Among 301 previously defined PIF-induced direct target genes[58], 51 were changed in *rcb-10*. Surprisingly, most of the RCB-dependent PIF-induced genes—35 of 51—were downregulated in *rcb-10* (Fig. 4f and Supplementary Data 3 and 4), suggesting that RCB promotes the expression of these PIF target genes. Most of the RCB-induced or RCB-repressed PIF target genes were also induced or repressed by HMR, respectively (Fig. 4g). We confirmed the results via qRT-PCR using select PIF-induced, RCB/HMR-dependent (Fig. 4h). Together, these results support a model in which RCB works in concert with HMR to promote the degradation of PIF1 and PIF3 and facilitate the expression of a subset of growth-relevant PIF target genes[57].

**PIFs repress PEP assembly in the dark**. PIFs inhibit chloroplast biogenesis in the dark by repressing nuclear-encoded photosynthesis genes[25,27]. However, it remains unclear whether PIFs also control the expression of *PhAPGs*. To test this possibility, we asked whether *PhAPG* expression is affected in the darkness in *pifq*, a quadruple *pif1/pif3/pif4/pif5* mutant[27]. Because etioplast differentiation is dependent on seedling development[7], we performed the experiments using Col-0 and *pifq* seedlings at different stages or days after seed germination. Strikingly, PEP-dependent *PhAPGs*, but not NEP-dependent genes, were upregulated by 2–11-fold in 3-d- and 4-d-old dark-grown *pifq* seedlings compared with those in Col-0 (Fig. 5a). *PhAPGs* were not upregulated in 2-d-old dark-grown *pifq* seedlings, suggesting that a PIF-independent developmental signal regulates *PhAPGs* in early seedling development. In agreement with *PhAPG* expression, 3-d- and 4-d-old, but not 2-d-old, dark-grown *pifq* seedlings could assemble the 1000-kDa PEP complex (Fig. 5b). Together, these results demonstrate that PIFs from the nucleus repress PEP assembly and *PhAPG* expression in the dark.

**RCB activates *PhAPG* expression primarily from the nucleus**. The dual localization of RCB raised the question of whether RCB regulates *PhAPG* expression directly in the plastids, from the nucleus, or in both. To address this question, we examined whether *rcb*'s defect in *PhAPG* expression was caused by the failure of PIF degradation in the nucleus. Surprisingly, knocking out *PIF1*, *PIF3*, *PIF4*, and *PIF5* in *rcb-10* rescued its long-hypocotyl and albino phenotypes (Fig. 6a, b). Moreover, the *rcb-10/pifq* mutant also rescued *rcb-10*'s defects in PEP assembly and *PhAPG* activation (Fig. 6c, d). These results demonstrate that RCB controls *PhAPG* expression primarily from the nucleus by facilitating PHY-mediated PIF degradation.

**RCB interacts with PHYB and promotes photobody biogenesis**. We next tested whether RCB interacts with PHYs. Supporting this hypothesis, PHYA and PHYB co-immunoprecipitated with RCB-HA-His (Fig. 7a). We used an in vitro GST pull-down assay to examine whether RCB interacts with PHYs directly. GST-RCB pulled down the PHYA-HA and PHYB-HA apoproteins, which lack a bilin chromophore (Fig. 7b). In addition, RCB could bind similarly to the Pfr and Pr forms of PHYA and PHYB (Fig. 7b). These results indicate that RCB interacts directly with PHYA and PHYB in a light-independent manner.

One of the earliest light responses at the cellular level is the localization of PHYB to photobodies[17]. Because the localization of PHYB to large photobodies is tightly associated with PIF3 degradation[22,28,29], we asked whether RCB is required for photobody biogenesis. In *PBG* seedlings grown under 10 μmol

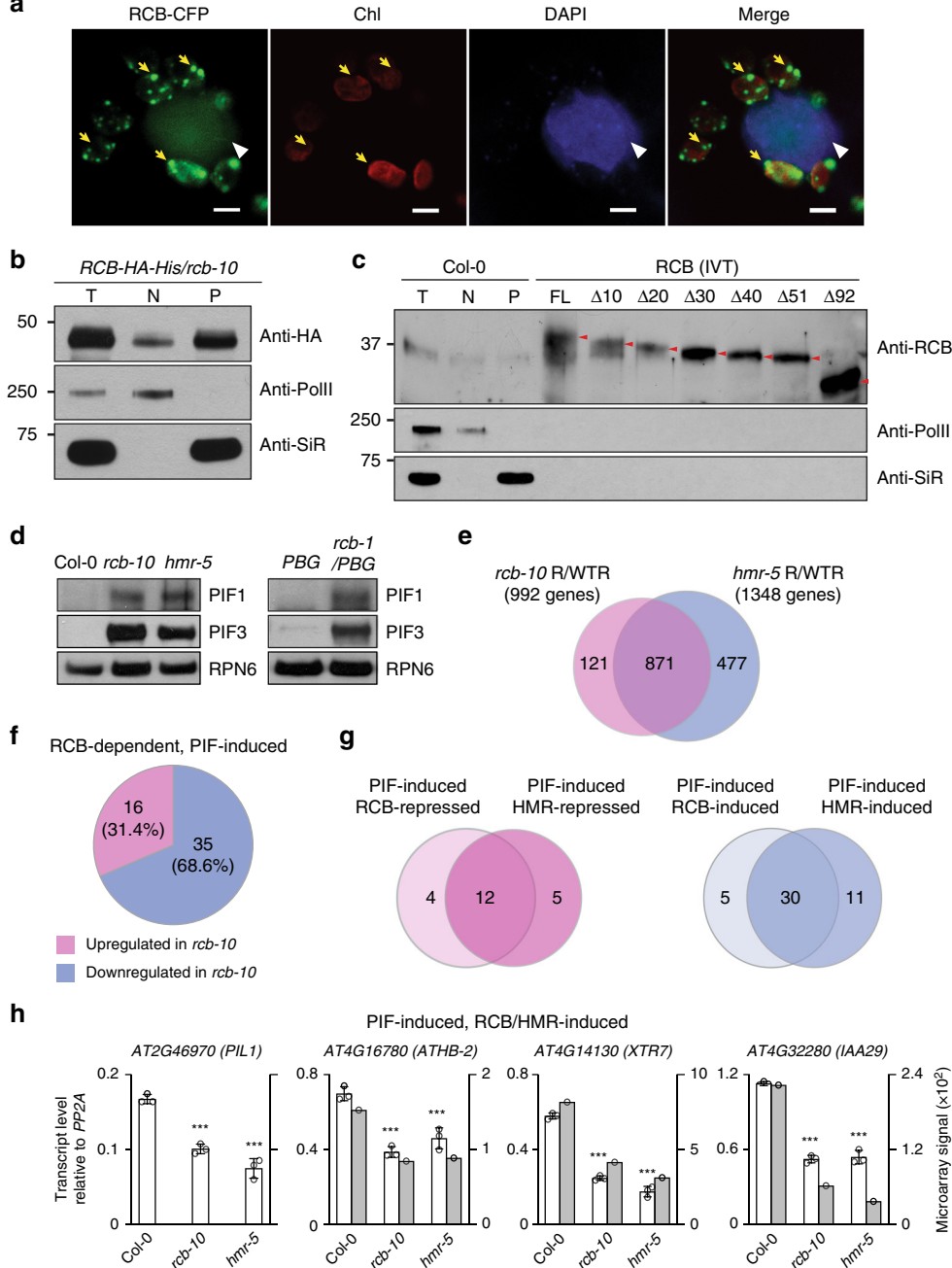

**Fig. 4** RCB is a dual-targeted protein that regulates the stability and activity of PIFs. **a** Confocal images showing that transiently-expressed RCB-CFP localizes to plastids and the nucleus in tobacco cells. Chlorophyll autofluorescence (Chl) and DAPI staining were used to label chloroplasts (indicated by yellow arrows) and nuclei (indicated by white arrowheads), respectively. The bars represent 5 μm. **b** Immunoblots showing total (T), nuclear (N), and plastidial (P) protein fractions from 4-d-old *RCB-HA-His/rcb-10* seedlings grown in 10 μmol m$^{-2}$ s$^{-1}$ R light. RCB-HA-His was detected using anti-HA antibodies. **c** Immunoblots showing endogenous RCB from total (T), nuclear (N), and plastidial (P) protein extracts from 4-d-old Col-0 seedlings grown in 10 μmol m$^{-2}$ s$^{-1}$ R light. RCB was detected using anti-RCB antibodies. In vitro translated (IVT) full-length RCB and a series of N-terminal truncation fragments were used as molecular size controls. For (**b**) and (**c**), RNA polymerase II (Pol II) and SiR were used as controls for the nuclear and plastidial fractions, respectively. **d** Immunoblots showing the levels of PIF1 and PIF3 in 4-d-old indicated lines grown in 10 μmol m$^{-2}$ s$^{-1}$ R light. RPN6 was used as a loading control. **e** Venn diagram showing that 871 of the 992 RCB-dependent genes overlap with the previously defined HMR-dependent genes. **f** Venn diagram showing that the majority of RCB-dependent, PIF-induced genes were downregulated in *rcb-10*. **g** Venn diagrams showing that RCB regulates similar PIF-regulated genes as HMR. **h** Microarray and qRT-PCR analyses of the mRNA levels of selected PIF-induced, RCB/HMR-induced genes in 4-d-old Col-0, *rcb-10*, and *hmr-5* seedlings grown in 10 μmol m$^{-2}$ s$^{-1}$ R light. The filled and open columns represent data from microarray and qRT-PCR, respectively. Transcript levels from the qRT-PCR experiments were calculated relative to those of *PP2A*. Error bars represent SD of three biological replicates. *** Indicates statistically significant differences in the transcript level from that of Col-0 (Student's *t*-test, $p \leq 0.001$). The source data underlying the immunoblots in (**b–d**) and the qRT-PCR analysis in (**h**) are provided in the Source Data file

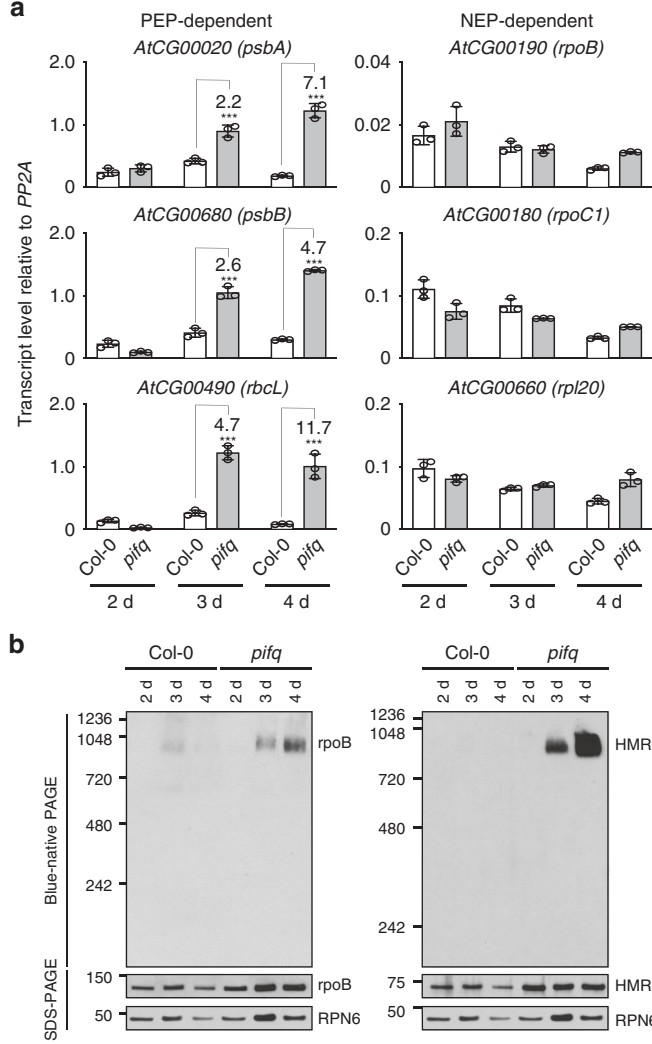

**Fig. 5** PIFs repress the assembly and activity of the PEP in the dark. **a** qRT-PCR results showing the transcript levels of representative PEP- and NEP-dependent genes in 2–4-day-old dark-grown Col-0 and *pifq* seedlings. The transcript levels were calculated relative to those of *PP2A*. Fold changes are shown only for the *pifq* samples exhibiting greater than twofold changes compared with the values of the corresponding Col-0 samples (***$p \leq$ 0.001). Error bars represent SD of three biological replicates. **b** Immunoblots showing the level of the PEP complex (blue-native PAGE) as well as the total amount of rpoB and HMR proteins (SDS-PAGE) in 2–4-day-old dark-grown Col-0 and *pifq* seedlings using antibodies against rpoB or HMR. RPN6 was used as a loading control. The source data underlying the qRT-PCR analysis in (**a**) and the immunoblots in (**b**) are provided in the Source Data file

m$^{-2}$ s$^{-1}$ R light, PHYB-GFP localized to a few large photobodies with a median volume of 2.56 μm$^3$ (Fig. 7c, d)[20]. In contrast, PHYB-GFP was found only in small photobodies with median volumes of 0.13 and 0.26 μm$^3$ in *rcb-1*/PBG and *rcb-2*/PBG, respectively (Fig. 7d). The average number of large photobodies with a volume >1.5 μm$^3$ decreased from 3.75 per nucleus in *PBG* to 0.15 and 0.7 per nucleus in *rcb-1*/PBG and *rcb-2*/PBG, respectively (Fig. 7e). While *PBG* nuclei had no photobodies smaller than 1.5 μm$^3$, there were an average of 69 and 33 small photobodies per nucleus in *rcb-1*/PBG and *rcb-2*/PBG, respectively (Fig. 7e). These results indicate that RCB functions to facilitate the localization of PHYB to large photobodies.

## Discussion

It has been postulated for decades that plastidial gene expression is determined by the developmental program of the host cell and therefore is ultimately controlled by the host cell's nucleus[5]. However, how the nucleus communicates with the plastids to coordinate plastidial gene expression with the cellular developmental program has been a long-standing question[5]. We have elucidated the framework of a nucleus-to-plastid or anterograde light signaling pathway that is initiated by PHYs in the nucleus to activate *PhAPG*s in the plastids for chloroplast biogenesis (Fig. 7f). Our results demonstrate that RCB-dependent photobody biogenesis and degradation of nuclear repressors of chloroplast biogenesis, PIF1 and PIF3, trigger anterograde signaling to the plastids for the assembly and activation of the PEP for *PhAPG* expression (Fig. 7f). Because PIFs are master repressors of photosynthesis-associated nuclear-encoded genes (*PhANG*s)[25,27], this signaling design enables the concerted activation of nuclear- and plastidial-encoded photosynthesis-associated genes during chloroplast biogenesis by one master switch—PHY-mediated PIF degradation in the nucleus (Fig. 7f).

Previous genetic screens failed to identify this anterograde signaling pathway because of both gene redundancy, as in the case of PIFs, as well as the lack of an effective strategy for identifying relevant mutants, as in the case of RCB (Fig. 7f). The forward genetic screening strategy for tall-and-albino mutants turned out to be successful in identifying missing PHY signaling components required for regulating plastidial transcription. The *hmr* mutant is the founding member of this mutant family[28]. HMR participates in nuclear PIF1 and PIF3 degradation and is a component of the PEP complex[28,37,57]. Thus, HMR also participates in both the nuclear and plastidial signaling events of the anterograde signaling (Fig. 1f)[57]. Consistent with HMR's dual functions in the nucleus-to-plastid signaling, the *hmr-5*/*pifq* mutant rescued only *hmr-5*'s long-hypocotyl phenotype but not its albino phenotype[57]. In contrast, *rcb-10*/*pifq* rescued both *rcb-10*'s long-hypocotyl and albino phenotypes (Fig 6), demonstrating that RCB acts primarily in the nuclear signaling event that mediates PIF1 and PIF3 degradation. The *rcb* mutant represents a unique subgroup of albino mutants whose albinism is not caused by defects in the chloroplast per se but rather due to the failure of degrading the nuclear repressors of chloroplast biogenesis (the PIFs) in the light.

The new model also predicts that PIF degradation either generates an agonistic signal promoting PEP assembly or eliminates an antagonistic signal repressing PEP assembly (Fig. 7f). One possibility is that the expression of one or some of the nuclear-encoded components of the PEP is regulated by PIFs. However, we did not observe significant changes in the expression of PEP-associated proteins and sigma factors between *rcb-10* and Col-0 and between *rcb-10* and *rcb-10*/*pifq* (Supplementary Fig. 4), suggesting that these factors might not be the signal. Of course, we still cannot rule out the possibility that these proteins are regulated by light at posttranscriptional levels. We expect that further biochemical characterization of HMR and RCB along with the study of other tall-and-albino mutants will help to identify the nucleus-to-plastid signal.

Our results demonstrate that in addition to their well-recognized roles as repressors of *PhANG*s, PIFs are master nuclear regulators that repress *PhAPG*s in the dark (Fig. 5). PIFs play overlapping roles in repressing photomorphogenesis in the dark by activating growth-relevant genes to promote hypocotyl elongation and by repressing nuclear-encoded photosynthesis genes to inhibit chloroplast biogenesis (Fig. 7f)[24,25,27]. In particular, PIF1, PIF3, and PIF5 negatively regulate chlorophyll biosynthesis and photosynthesis genes[25,55,56]. Here we show that the *rcb* mutants, which accumulated PIF1 and PIF3 in the light

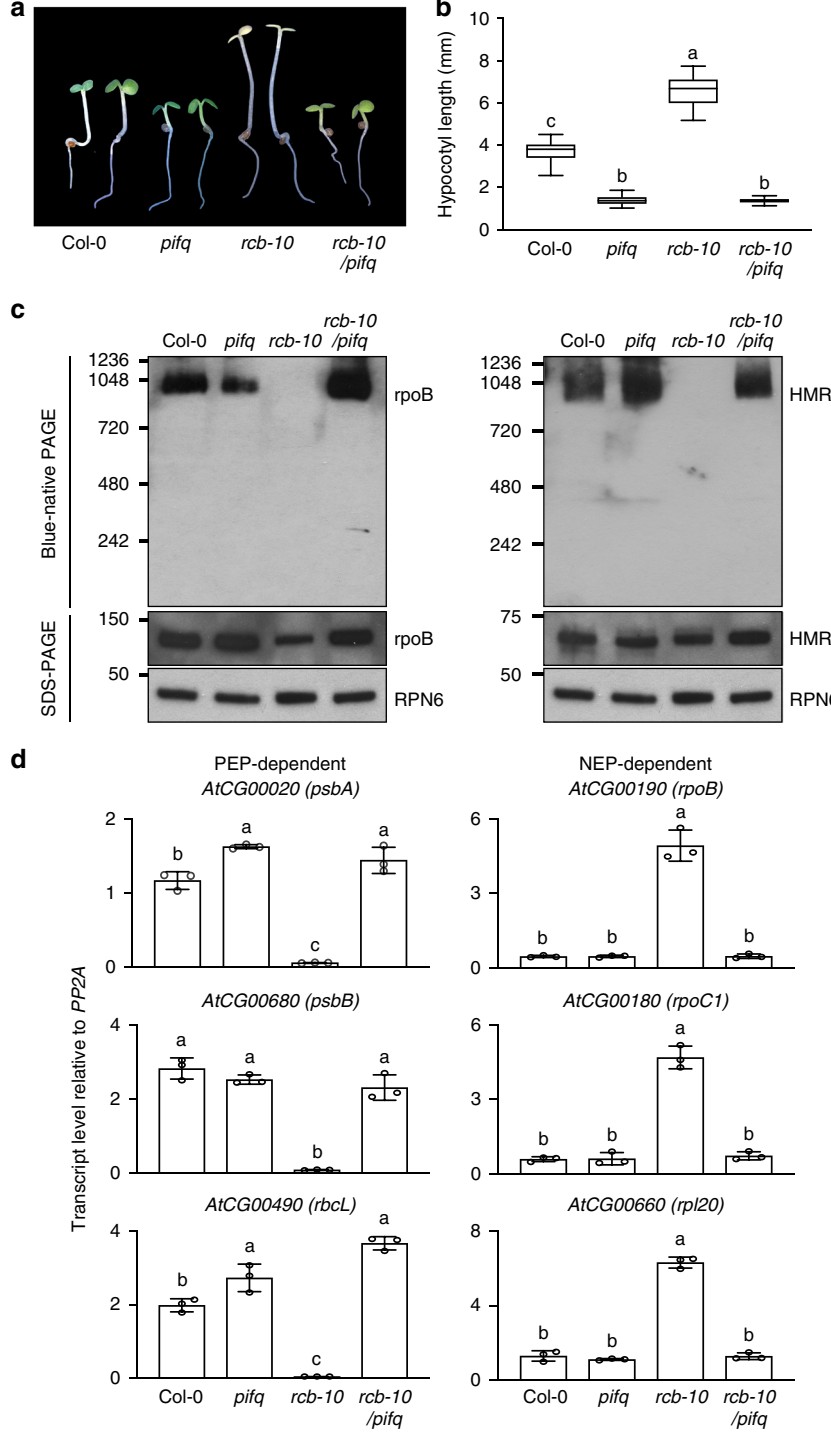

**Fig. 6** Removal of four *PIF*s in *rcb-10* rescued its tall and albino phenotypes. **a** Representative images of 4-d-old Col-0, *pifq*, *rcb-10*, and *rcb-10/pifq* seedlings grown in 10 µmol m$^{-2}$ s$^{-1}$ R light. **b** Box-and-whisker plots of hypocotyl length measurements for the seedlings shown in (**a**). The boxes represent from the 25th to the 75th percentiles, and the bars equal the median values. Different letters denote statistically significant differences in hypocotyl length (ANOVA, Tukey's HSD, $p \leq 0.001$). **c** Immunoblots showing the level of the PEP complex (blue-native PAGE), as well as the total amount of rpoB and HMR (SDS-PAGE) in 4-d-old Col-0, *pifq*, *rcb-10*, and *rcb-10/pifq* seedlings grown in 10 µmol m$^{-2}$ s$^{-1}$ R light. RPN6 was used as a loading control. **d**, qRT-PCR analyses of the transcript levels of representative PEP- and NEP-dependent genes in 4-d-old Col-0, *pifq*, *rcb-10*, and *rcb-10/pifq* seedlings grown in 10 µmol m$^{-2}$ s$^{-1}$ R light. Different letters denote statistically significant differences in hypocotyl length (ANOVA, Tukey's HSD, $p \leq 0.001$). Error bars represent SD of three biological replicates. The source data underlying the hypocotyl measurements in (**b**), the immunoblots in (**c**), and the qRT-PCR analysis in (**d**) are provided in the Source Data file

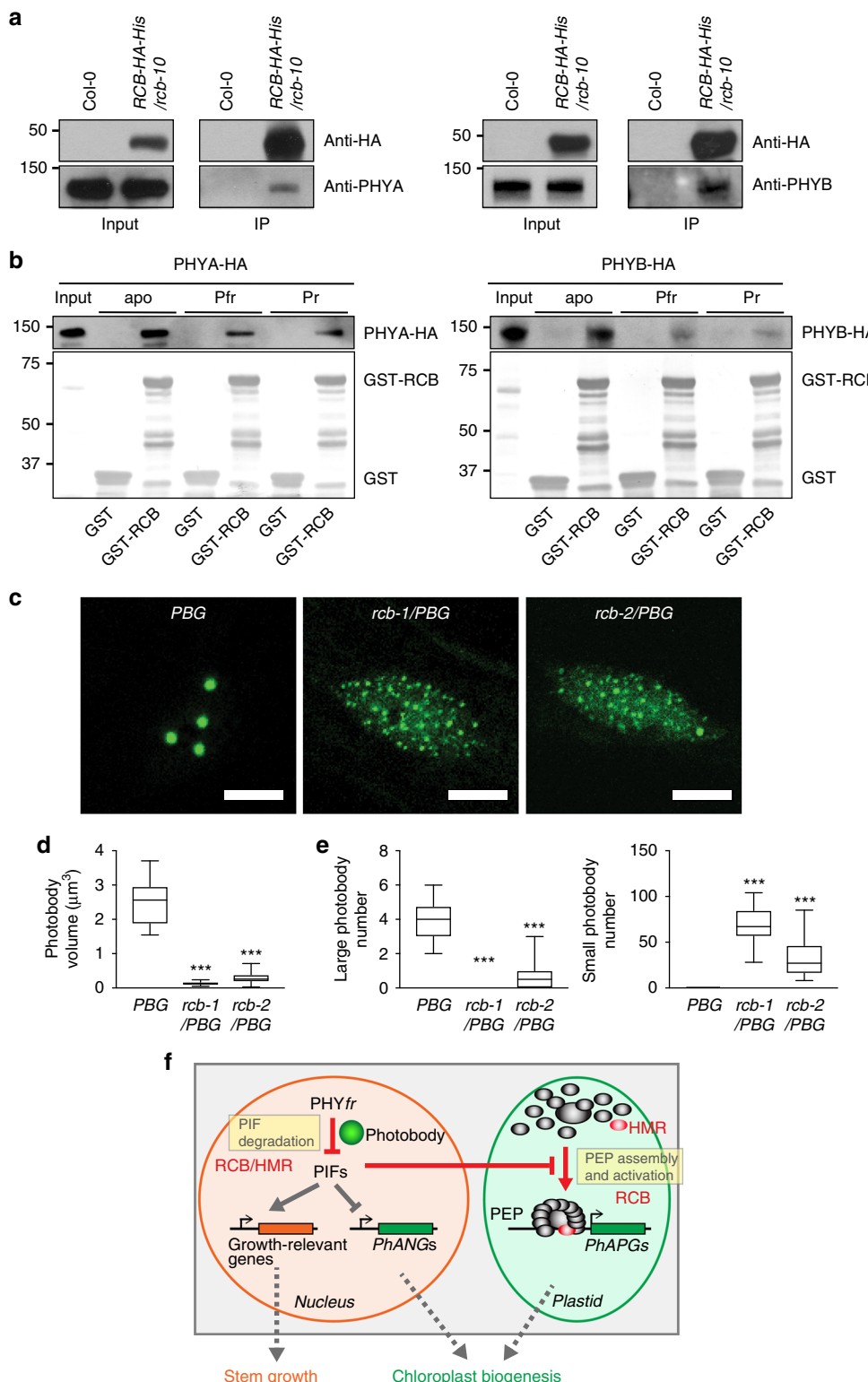

(Fig. 4d), failed to stimulate PEP assembly and *PhAPG* expression in the light (Fig. 2g, f). Both PEP assembly and *PhAPG* expression were de-repressed in the *pifq* mutant in the dark (Fig. 5) as well as in *rcb-10/pifq* in the light (Fig. 6c, d), indicating that PIFs are also nuclear repressors of *PhAPG* expression. It is worth noting that it remains unclear whether all PIFs are required for repressing *PhAPG* expression, which requires further genetic analysis of the combinations of triple *pif* mutants in the future. Nonetheless, our results demonstrate that PHY- and RCB-mediated PIF degradation is a central nuclear event that triggers *PhAPG* expression in plastids.

The mechanisms of PIF degradation have been extensively studied, particularly for PIF3, the founding member of the PIF family[23]. PIF3 interacts preferentially with the active Pfr forms of

**Fig. 7** RCB interacts directly with PHYs to promote photobody biogenesis. **a** RCB interacts with PHYA and PHYB in vivo. RCB-HA-His was immunoprecipitated using an anti-HA affinity matrix from 4-d-old *RCB-HA-His/rcb-10* seedlings grown in either 0.5 µmol m$^{-2}$ s$^{-1}$ FR light for the PHYA pull-down experiments or 10 µmol m$^{-2}$ s$^{-1}$ R light for the PHYB pull-down experiments. Samples from Col-0 seedlings grown under the same conditions were used as negative controls. Input and immunoprecipitated fractions of RCB-HA-His and PHYA (left panels) or PHYB (right panels) are shown. **b** RCB interacts directly with PHYA and PHYB. GST pull-down assays using *E. coli*-expressed GST-RCB or GST to pull-down in vitro translated HA-tagged PHYA (left panel) and PHYB (right panel). The pull-down assays were carried out with PHY apoproteins (apo) or holoproteins in either the Pr or Pfr form. The upper panels show immunoblots of input and bound PHY fractions using anti-PHYA or anti-PHYB antibodies. Immobilized GST and GST-RCB are shown in the Coomassie blue-stained SDS-PAGE gel in the bottom panels. **c** Representative confocal images of PHYB-GFP localization in 4-d-old *PBG*, *rcb-1/PBG*, and *rcb-2/PBG* seedlings grown in 10 µmol m$^{-2}$ s$^{-1}$ R light. The bars represent 5 µm. **d** Box-and-whisker plots showing the volumes of photobodies in *PBG*, *rcb-1/PBG*, and *rcb-2/PBG* seedlings grown in 10 µmol m$^{-2}$ s$^{-1}$ R light. **e** Box-and-whisker plots showing the numbers of large (≥1.5 µm$^3$) and small (<1.5 µm$^3$) photobodies in *PBG*, *rcb-1/PBG*, and *rcb-2/PBG* seedlings grown in 10 µmol m$^{-2}$ s$^{-1}$ R light. For (**d**) and (**e**), the boxes represent from the 25th to the 75th percentiles, and the bars equal the median values. *** Indicates statistically significant differences from that of *PBG* (Student *t*-test, $p \leq 0.001$). **f** Model of nucleus-to-plastid anterograde signaling for initiating chloroplast biogenesis. The source data underlying the immunoblots in (**a**) and (**b**) and the photobody analysis in (**d**) and (**e**) are provided in the Source Data file

PHYA and PHYB[59]. Although PIF3 interacts with both the N- and C-terminal modules of PHYB[60], the interaction with the C-terminal output module of PHYB is required for its degradation[29]. PHYB promotes PIF3 phosphorylation by PPK1-4 (photoregulatory protein kinase 1-4) and subsequent PIF3 degradation in the light by the ubiquitin–proteasome pathway[23,61]. At the cellular level, PIF3 degradation is closely associated with its localization to photobodies[22,23]. HMR, an acidic-type transcriptional activator required for photobody biogenesis, can interact directly with all PIFs and is required for PIF3 degradation as well as the activation of distinct set of growth-relevant genes[28,57,62]. We have proposed that PIF3 degradation is coupled to the transcriptional activation of the growth-relevant genes through HMR's transactivation domain[57]. The results from this study agree with this hypothesis. Similar to HMR, RCB is required for PIF1 and PIF3 degradation in the light (Fig. 4d). Global gene expression analyses indicate that RCB and HMR work in concert to regulate a similar set of genes (Fig. 4e–h). RCB regulates the majority of HMR-dependent PIF target genes in the same manner as HMR; specifically, RCB is required for the expression of HMR-dependent growth-relevant PIF target genes (Fig. 4g, h)[57]. Moreover, RCB interacts directly with PHYA and PHYB and is required for the localization of PHYB to large photobodies (Fig. 7a, b). These results provide evidence supporting the model that RCB works with HMR to regulate PIF degradation and transcriptional activity by facilitating photobody biogenesis (Fig. 7f).

We provide evidence that a critical mechanism by which light activates the PEP is by promoting its assembly into a 1000-kDa protein complex (Fig. 1). Extensive proteomic studies of the PEP complex in *Arabidopsis* and mustard have identified twelve essential PEP-associated proteins[1,37,38]. Maize PEP also forms a 1000-kDa complex[40,63]. Our results show that in *Arabidopsis*, the assembly of the 1000-kDa PEP complex is light-dependent (Fig. 1g, h). The PEP in mustard (*Sinapis alba* L.) seedlings has also been shown to form distinct light-dependent complexes: a larger 700-kDa complex enriched in chloroplasts in the light and a 420-kDa complex, likely made of the bacterial-like core subunits without (most of) the plant accessory proteins, enriched in non-photosynthetic etioplasts in the dark[64]. Both complexes are transcriptionally active[64,65]. However, we did not observe a smaller PEP complex in *Arabidopsis*. It is well known that the core subunits of the bacterial RNA polymerase is self-sufficient to form a functional complex[41]. Our results suggest that the *Arabidopsis* PEP complex requires additional plant-specific associated protein factors for its assembly. The formation of the 1000-kDa PEP complex in *Arabidopsis* is promoted by PHYs and RCB and repressed by PIFs (Figs 1g, 1h, 2g, 5b, 6c) and correlates with

*PhAPG* expression (Figs 1e, 1f, 2f, 5a, 6d). These results indicate that PEP assembly is a critical switching mechanism that activates *PhAPG* expression (Fig. 7f).

Our results demonstrate that RCB is a dual-targeted nuclear/plastidial protein. RCB was thought to localize exclusively to plastids[49–52]. In contrast, our results show that RCB is also targeted to the nucleus (Fig. 4a–c). Although we provide genetic evidence demonstrating that RCB triggers *PhAPG* expression mainly from the nucleus by facilitating PIF degradation (Fig. 7f), we do not exclude a direct function for RCB in plastids, where RCB is associated with the PEP complex in the nucleoid[51] and is required for the maintenance of the photosynthetic apparatus[66]. It is possible that the plastidial function of RCB is not essential for the activity of the PEP or that knocking out the *PIFs* in *rcb-10* can bypass these plastidial functions of RCB to trigger chloroplast biogenesis. Further investigations are needed to define precisely the function of RCB in plastids and to interrogate the role of dual localization of RCB in nucleus-and-plastid communication[2,67].

## Methods

**Plant materials and growth conditions.** The *Arabidopsis* mutants *phyB-9*[14], *phyA-211*[68], *phyA-211/phyB-9*[68], *phyABCDE*[11], *YHB*[36], and *pifq*[24] in the Columbia (Col-0) background were used for the characterization of plastidial gene expression and the assembly of the PEP complex. The *PBG* and *YHB* lines in the Landsberg *erecta* (L*er*) background have been previously reported[17,36]. Additional *Arabidopsis* transgenic lines and other mutants generated in this study are described below. Seeds were surface-sterilized and plated on half-strength Murashige and Skoog (MS) media with Gamborg's vitamins (MSP0506, Caisson Laboratories, North Logan, UT), 0.5 mM MES pH 5.7, and 0.8% agar (w/v) (A038, Caisson Laboratories, North Logan, UT)[62]. Seeds were stratified in the dark at 4 °C for 4 days before being placed in an LED chamber (Percival Scientific, Perry, IA) in the indicated light conditions. The fluence rate of light was determined using an Apogee PS200 spectroradiometer (Apogee Instruments Inc., Logan, UT) and SpectraWiz Software (StellarNet, Tampa, FL). Representative seedlings in the indicated light conditions were imaged using a Leica MZ FLIII stereo microscope (Leica Microsystems Inc., Buffalo Grove, IL) and the images were processed using Adobe Photoshop CC (Adobe Systems, Mountain View, CA).

**Chlorophyll measurement.** Total chlorophyll from 100 mg of seedlings of the indicated genotypes and growth conditions was extracted in 3 ml of 100% dimethyl sulphoxide (DMSO) with incubation at 65 °C for 30 min. Then, the absorbance at 648.2 and 664.9 nm was measured by spectrophotometry. The concentrations of total chlorophyll were calculated using equation ($C_{a+b} = 7.49A^{664.9} + 20.34A^{648.2}$) derived from the specific absorption coefficients for chlorophylls *a* and *b* in 100% DMSO[69].

**RNA extraction and quantitative real-time PCR.** Total RNA from seedlings of the indicated genotypes and growth conditions was isolated using a Spectrum Plant Total RNA Kit (Sigma-Aldrich, St. Louis, MO) or Quick-RNA MiniPrep Kit with on-column DNase I treatment (Zymo Research, Irvine, CA). cDNA was synthesized from total RNA using a Superscript II First-Strand cDNA Synthesis Kit (Thermo Fisher Scientific, Waltham, MA) according to the manufacturer's protocol. Oligo(dT) primers were used for the analysis of nuclear gene expression, and

a mixture of plastidial-gene-specific primers was used for the analysis of plastidial genes. qRT-PCR was performed with FastStart Universal SYBR Green Master Mix on a LightCycler 96 System (Roche, Basel, Switzerland). The mRNA level of gene-of-interest was normalized to that of *PP2A*. Primers for qRT-PCR and cDNA synthesis are listed in Supplementary Tables 1 and 2.

**Blue-native gel electrophoresis analysis of PEP assembly**. PEP assembly was analyzed by blue-native polyacrylamide gel electrophoresis (blue-native PAGE) using a NativePAGE Sample Prep Kit and a NativePAGE Novex Bis-Tris Gel system (Thermo Fisher Scientific, Waltham, MA) with immunoblot[63,70]. Seedlings grown under the indicated conditions were ground in liquid nitrogen and resuspended in three volumes of BN lysis buffer (100 mM Tris-Cl, pH 7.2; 10 mM MgCl$_2$; 25% glycerol; 1% Triton X-100; 10 mM NaF; 5 mM β-mercaptoethanol; 1 × EDTA-free protease inhibitor cocktail). Protein extracts were divided into two tubes, one for blue-native PAGE and the other for SDS-PAGE. For blue-native PAGE, protein extracts were mixed with BN sample buffer (1 × NativePAGE sample buffer, 50 mM 6-aminocaproic acid, 1% n-dodecyl β-D-maltoside, and Benzonase nuclease) and incubated for 60 min at room temperature to degrade DNA/RNA and solubilize the PEP complex. Samples were mixed with 0.25% NativePAGE Coomassie blue G-250 sample additive and centrifuged at 17,500 × g for 10 min at 4 °C. Proteins from the supernatant were separated on a 4–16% NativePAGE Bis-Tris protein gel using a NativePAGE Running Buffer Kit (Thermo Fisher Scientific, Waltham, MA) according to the manufacturer's protocol and with the following modifications. NativeMark Unstained Protein Standard (Thermo Fisher Scientific, Waltham, MA) was used to determine protein size in blue-native PAGE. Briefly, electrophoresis was performed at a constant 30~40 V for 3 h at 4 °C until the blue dye migrated through one-third (1/3) of the gel. The Dark Blue Cathode Buffer was replaced with Light Blue Cathode Buffer, and electrophoresis was continued at a constant 20~25 V overnight at 4 °C. After electrophoresis was complete, the separated proteins were transferred onto a polyvinylidene difluoride (PVDF) membrane using 1 × NuPAGE Transfer Buffer (Thermo Fisher Scientific, Waltham, MA) at a constant 70 V for 7 h at 4 °C. After transfer, the membrane was fixed with fixation buffer (25% methanol, 10% acetic acid) for 15 min and washed with water. The membrane was incubated with methanol for 3 min to destain the dye and washed with water, and then immunoblotting continued. The membrane was blocked with 2% non-fat milk in 1 × TBS (20 mM Tris-Cl pH 7.6, 150 mM NaCl), probed with the indicated monoclonal mouse anti-rpoB antibodies (PHY1700, PhytoAB Inc., Redwood City, CA) or polyclonal rabbit anti-HMR antibodies[28], washed with 1 × TBS containing 0.05% Tween-20, and then incubated with secondary antibodies conjugated with horseradish peroxidase.

**Protein extraction and immunoblot analysis**. Total protein was extracted from *Arabidopsis* seedlings grown under the indicated conditions. Plant tissues were ground in liquid nitrogen and resuspended in extraction buffer (100 mM Tris-HCl pH 7.5, 100 mM NaCl, 1% SDS, 5 mM EDTA pH 8.0, 20 mM DTT, 40 μM MG132, 40 μM MG115, and 1 × EDTA-free protease inhibitor cocktail)[62]. For PIF1 and PIF3 protein analyses, plant tissues were directly ground in extraction buffer containing 100 mM Tris-HCl pH 7.5, 100 mM NaCl, 5% SDS, 5 mM EDTA pH 8.0, 20% glycerol, 20 mM DTT, 40 mM β-mercaptoethanol, 2 mM PMSF, 1 mM bromophenol blue, 1% phosphatase inhibitor cocktail 3, 80 μM MG132, 80 μM MG115, and 1 × EDTA-free protease inhibitor cocktail in a 1:3 (mg/μl) ratio at room temperature, boiled for 10 min and then centrifuged at 20,000 × g for 10 min at room temperature. Protein extracts were separated via SDS-PAGE, transferred to PVDF membranes, probed with the indicated primary antibodies, and then incubated with HRP-conjugated secondary antibodies. Rabbit polyclonal anti-PIF1[28], rabbit polyclonal anti-PIF3[28], mouse monoclonal anti-PHYA and anti-PHYB (gift from Dr. Akira Nagatani), mouse monoclonal anti-HA (H3663, Sigma-Aldrich, St. Louis, MO), goat polyclonal anti-HA (A00168, Genscript, Piscataway, NJ), and anti-RPN6 (BML-PW8370-0100, Enzo Life Sciences, Farmingdale, NY) antibodies were used at a 1:1000 dilution. Anti-mouse (1706516, Bio-Rad, Hercules, CA), anti-rabbit (1706515, Bio-Rad, Hercules, CA), and anti-goat (sc-2020, Santa Cruz Biotechnology, Santa Cruz, CA) secondary antibodies were used at a 1:5000 dilution. Signals were detected via SuperSignal West Dura Extended Duration Chemiluminescent Substrate (Thermo Fisher Scientific, Waltham, MA).

**Mutant generation**. *PBG* seeds (0.2 g) were hydrated in 45 ml of ddH$_2$O with 0.005% Tween-20 and left on a tube rotator for 4 h. The seeds were washed with ddH$_2$O twice and then soaked in 1 mM N-ethyl-N-nitrosourea or 0.2% ethyl methanesulfonate (Sigma-Aldrich, St. Louis, MO) solutions for 15 h with rotation. Then, the seeds were washed with ddH$_2$O eight times, stratified in the dark at 4 °C for 4 days, and then sown onto large plates. The seedlings (M1 generation) were transferred to soil, and the progeny (M$_2$ generation) were collected from individual plants.

**Genetic mapping by SHOREmap**. *rcb-1/PBG* and *rcb-2/PBG* mutants in the Landsberg background were crossed to the Col-0 reference accession of *A. thaliana*. Genomic DNA from pools of 800 F$_2$ seedlings with a tall-and-albino phenotype was extracted as follows[71]. Seedlings were ground in liquid nitrogen and

resuspended in nuclei extraction buffer (10 mM Tris-HCl pH 9.5, 10 mM EDTA pH 8.0, 100 mM KCl, 500 mM sucrose, 4 mM spermidine, 1 mM spermine, 0.1% β-mercaptoethanol). After the homogenized tissues were filtered through Miracloth (Calbiochem), 2 ml of lysis buffer (10% Triton X-100 in nuclei extraction buffer) was added and kept on ice for 2 min. The homogenate was centrifuged at 2000 × g for 10 min at 4 °C. The nuclei pellet was resuspended in 500 μl of CTAB extraction buffer (100 mM Tris-HCl pH 7.5, 0.7 M NaCl, 10 mM EDTA pH 8.0, 1% CTAB, 1% β-mercaptoethanol) and incubated for 30 min at 60 °C. Genomic DNA was extracted with chloroform/isopentanol (24:1) and precipitated with isopropanol with centrifugation at 20,000 × g for 10 min at 4 °C. DNA pellet was washed with 75% ethanol and resuspended in water. Illumina paired-end libraries with 300-bp insert sizes were constructed for both the *rcb-1/PBG* and *rcb-2/PBG* pools following the manufacturer's instructions. Eighty-base-pair paired-end reads were generated on an Illumina Genome Analyzer II (Illumina, San Diego, CA), targeting ~25 × genome coverage. SNPs, indels up to 3 bp, and large deletions were called with SHORE[72]. Genomic regions enriched for mutant parental markers were identified with SHOREmap[73,74]. Variants in the final mapping interval that were absent from the Landsberg background and that were predicted to have a large impact on ORF integrity were prioritized as candidate mutations. The *rcb-1* allele was genotyped with a dCAPS (derived cleaved amplified polymorphic sequences) marker using the PCR primers rcb1-F (gtatttcgatacacatccaaccaaaa) and rcb-R (gttctactatgcacaccaag); XcmI digestion of the PCR product gave one 231-bp fragment for Col-0 and two fragments of 203-bp and 28-bp for *rcb-1*. The *rcb-2* allele was genotyped with a dCAPS marker using the PCR primers rcb2-F (ggagatgatgggagtgagattgctt) and rcb-R; DdeI digestion of the PCR product yielded one 197-bp fragment for Col-0 and two fragments of 174-bp and 23-bp for *rcb-2*.

**Constructs and transgenic plants**. The primers used to generate the plasmid constructs are listed in Supplementary Table 3. The cDNA of At4g28590 was amplified by PCR and ligated into the SacI and KpnI sites of *pCHF1*[75] to express RCB driven by the constitutive cauliflower mosaic virus 35S promoter. The *RCB-HA-His* construct was made by subcloning full-length *RCB* cDNA without a stop codon into the SacI and KpnI sites of the *pCHF1-(PT)4P-HA-His* vector, which was prepared by inserting a DNA fragment encoding (PT)4P-3×HA-6×His into the SalI and PstI sites of the *pCHF1* vector. Transgenic lines expressing At4g28590 or RCB-HA-His constructs were generated by transforming *rcb-1/PBG* or *rcb-10* plants with *Agrobacterium tumefaciens* strain GV3101 harboring these constructs. Multiple independent lines from the T1 generation were selected on MS medium containing 100 μg/ml gentamicin. T1 plants with homozygous *rcb-1* mutations were selected, and the single-locus-insertion status of the transgene in the T2 generation was determined based on a 3:1 segregation ratio for gentamycin resistance. T3-generation plants homozygous for the transgene were used for the further experiments.

The *GST-RCB* construct used for GST pull-down assays was generated by amplifying the cDNA of RCB and ligating it into the BamHI and PstI sites of *pET42b* (EMD Biosciences, San Diego, CA) using T4 DNA ligase (New England Biolabs, Ipswich, MA). The *pET42b* empty vector or *pET42b-RCB* was transformed into *E. coli* BL21 (DE3) cells. Lysates from bacteria expressing recombinant GST or GST-RCB proteins were prepared for GST pull-down assays.

The *RCB-CFP* used for the tobacco transient assay was generated by amplifying the cDNA of RCB and ligating it into the *pCHF1-CFP* vector. The *pCHF1-CFP* vector was generated by subcloning the coding sequence of *CFP* into the SalI and PstI sites of the *pCHF1* vector. The *RCB-CFP* plasmid was transformed into *Agrobacterium tumefaciens* strain GV3101.

The full-length RCB and N-terminally truncated RCB fragments were generated by amplifying a series of *RCB* cDNAs and ligating them into the PstI and BamHI sites of the *pCMX-PL2* vector.

**Hypocotyl measurement**. For hypocotyl length measurement, 4-d-old seedlings grown under different light conditions were scanned using an Epson Perfection V700 photo scanner, and hypocotyls were measured using NIH imageJ software (https://imagej.nih.gov/ij/). Box-and-whisker plots of hypocotyl measurements were drawn using Prism 7 software (GraphPad software, Inc., La Jolla, CA).

**Confocal imaging and photobody analysis**. Photobody analysis of 4-d-old *PBG*, *rcb-1/PBG*, and *rcb-2/PBG* mutants was performed on a Zeiss LSM 510 inverted confocal microscope using a 100 × HCX PL APO oil immersion objective (Carl Zeiss, Jena, Germany). GFP signals were detected with 488-nm excitation from an argon laser and 505- to 550-nm bandpass detector settings. Images were collected using LSM510 software version 4.2. The volume and number of photobodies were analyzed using Huygens Essential Software (Scientific Volume Imaging, Hilversum, Netherlands)[21]. The object analyzer tool was used to threshold the images and analyze the volume and number of photobodies.

Transient expression assays for RCB-CFP localization were performed using *Nicotiana benthamiana*[76]. The *Agrobacterium tumefaciens* strain GV3101 harboring was grown overnight in 4 ml of LB media and pelleted by centrifugation at 3,000 × g. The bacterial pellet was resuspended in 2 ml of infiltration buffer (10 mM MES pH 5.7, 10 mM MgCl$_2$) with 200 μM acetosyringone and incubated for 2 h at room temperature. The bacterial suspension was diluted to OD$_{600}$ of 1.0 with infiltration buffer and infiltrated into the abaxial side of leaves from

3-week-old tobacco. Seventy-two hours after infiltration, leaf punches were mounted in PBS and imaged on a Zeiss LSM 510 inverted confocal microscope with a 40 × Plan-Apochromat dry objective (Carl Zeiss, Jena, Germany). DAPI was monitored using excitation from a 405 nm diode laser and a 420–480-nm bandpass detector, chlorophyll was monitored using excitation from a 405 nm diode laser and a 615-nm longpass detector, and CFP was monitored using 458 nm excitation from an argon laser and a 470–500-nm bandpass detector.

**Generation of polyclonal anti-RCB antibody**. A fragment of RCB encoding the central region (amino acids 93–221) was amplified via PCR using primers listed in Supplementary Table 3 and cloned into the BamHI and PstI sites of pET42b (EMD Biosciences, San Diego, CA). The recombinant proteins, affinity-purified using glutathione sepharose 4B beads (GE Healthcare, Chicago, IL), were separated via SDS-PAGE. Bands of interest were excised and used as antigens for antibody production. Antibodies were produced in rabbits by Cocalico Biologicals, Inc. (Reamstown, PA).

**Nuclear and plastid fractionation**. Seedlings grown in the indicated light conditions were frozen in liquid nitrogen and ground using a mortar and pestle. The ground tissue was homogenized with nuclei extraction buffer (2 ml per gram fresh weight) containing 20 mM PIPES pH 6.5, 10 mM MgCl$_2$, 12% hexylene glycol, 0.25% Triton X-100, 4 mM β-mercaptoethanol, 40 μM MG115/MG132, and 1 × EDTA-free protease inhibitor cocktail. The lysate was filtered through Miracloth, loaded on top of 2 mL of a 30% Percoll solution, and centrifuged at 700 × g for 5 min at 4 °C. The nuclear pellet was resuspended in nuclear extraction buffer, loaded onto 30% Percoll solution, and centrifuged at 700 × g to remove any remaining chlorophyll contamination from the nuclear pellet. The final nuclear pellet was dissolved in buffer containing 100 mM Tris-HCl pH 7.5, 100 mM NaCl, 1% SDS, 5 mM EDTA, 5 mM DTT, 10 mM β-mercaptoethanol, 40 μM MG115/MG132, and 1 × EDTA-free protease inhibitor cocktail.

For chloroplast fractionation, 1 g of seedlings was homogenized with 2 ml of grinding buffer (50 mM HEPES-KOH pH 7.3, 0.33 M sorbitol, 0.1% BSA, 1 mM MnCl$_2$, 2 mM EDTA)[28]. The homogenized extract was filtered through Miracloth and centrifuged for 5 min at 2600 × g to precipitate chloroplasts. The crude chloroplasts resuspended in grinding buffer were further fractionated using a stepwise gradient of 80 and 40% Percoll. Intact chloroplasts were isolated from the boundary layer between the 80 and 40% Percoll.

**Microarray analysis**. Col-0 and *rcb-10* seedlings grown in 10 μmol m$^{-2}$ s$^{-1}$ continuous R light for 4 days were used for microarray analysis. Three independent biological replicates of each genotype were grown, sampled separately, and used to extract total RNA using a Spectrum Plant Total RNA Kit (Sigma-Aldrich, St. Louis, MO). Total RNA was assessed for quality with an Agilent 2100 Bioanalyzer G2939A (Agilent Technologies, Santa Clara, CA) and a Nanodrop 8000 spectrophotometer (Thermo Fisher Scientific, Waltham, MA). Hybridization targets were prepared from total RNA with a MessageAmp$^{TM}$ Premier RNA Amplification Kit (Thermo Fisher Scientific, Waltham, MA), hybridized to GeneChip® ATH1 Genome arrays in an Affymetrix GeneChip® hybridization oven 645, washed in an Affymetrix GeneChip® Fluidics Station 450 and scanned with an Affymetrix GeneChip® Scanner 7G according to standard Affymetrix GeneChip® hybridization, wash, and stain protocols (Affymetrix, Santa Clara, CA).

The microarray data analysis was performed using GeneSpring GX software version 12.1 (Agilent Technologies, Santa Clara, CA). The *cel* files were normalized using the robust multiarray average (RMA) background correction method. Raw and normalized data have been deposited in the GEO repository under accession number GSE122351. Volcano plot filtering was applied for pairwise comparisons between Col-0 and *rcb-10*. Significantly differentially expressed genes were selected with the following thresholds: corrected *p*-value <0.05 (adjusted with Benjamini-Hochberg FDR) and absolute fold change ≥2.0.

**Co-immunoprecipitation**. One gram of 4-d-old *RCB-HA-HIS/rcb-10* seedlings grown under the indicated light condition was ground in liquid nitrogen and resuspended in 2 ml of Co-IP buffer (50 mM Tris-HCl pH 7.5, 100 mM NaCl, 1 mM EDTA, 2 mM DTT, 0.1% IGEPAL CA-630, 40 μM MG115/MG132, and 1 × EDTA-free protease inhibitor cocktail)[62]. The crude extract was cleared by centrifugation at 20,000 × g for 20 min at 4 °C and filtered through a 0.45-μm polyethersulfone filter (VWR International, Radnor, PA). One hundred microliters of anti-HA affinity matrix (Sigma-Aldrich, St. Louis, MO) was added into the cleared extract, incubated for 4 h at 4 °C, and then washed four times with Co-IP buffer. The immunoprecipitated proteins were eluted with standard SDS Laemmli buffer and used for SDS-PAGE and immunoblot.

**GST pulldown assay**. GST-fusion proteins were expressed in *Escherichia coli* BL21 (DE3) carrying pET42b vectors. Bacterial cells were pelleted by centrifugation at 5000 × g for 10 min at 4 °C and resuspended in E buffer (50 mM Tris-HCl pH 7.5, 100 mM NaCl, 1 mM EDTA, 1 mM EGTA, 1% DMSO, 2 mM DTT, and bacterial protease inhibitor cocktail)[62]. The cells were lysed by French press and centrifuged at 20,000 × g for 20 min at 4 °C. The proteins were precipitated with ammonium sulfate (0.46 g/ml) for 4 h at 4 °C and centrifuged at 10,000 × g for 20 min at 4 °C. The lysates were cleared by ultracentrifugation at 50,000 × g for 15 min at 4 °C. The

cleared lysates were incubated with glutathione sepharose beads (GE Healthcare, Chicago, IL) in E buffer for 2 h at 4 °C.

Prey constructs for the apoproteins of PHYA and PHYB (*pCMX-PL2-PHYA* and *pCMX-PL2-PHYB*) were prepared using a TNT T7 Coupled Reticulocyte Lysate system (Promega, Madison, WI) according to the manufacturer's protocol[62]. Holoproteins of PHYA and PHYB were generated by incubating the apoproteins with 20 μM phycocyanobilin (PCB) for 1 h in the dark on ice. The in vitro-translated proteins were diluted with E buffer containing 0.1% IGEPAL CA-630 and further incubated with the immobilized GST-fusion protein for 2 h. The beads were washed four times with E buffer + 0.1% IGEPAL CA-630. Bound proteins were eluted by boiling in standard SDS Laemmli buffer.

**Reporting summary**. Further information on research design is available in the Nature Research Reporting Summary linked to this article.

## Data availability

*Arabidopsis* mutants and transgenic lines, as well as plasmids generated during the current study, are available from the corresponding author upon reasonable request. The microarray data for RCB-dependent gene expression have been deposited in the GEO repository under accession number GSE122351. The source data underlying Figs. 1b, 1d-h, 2b, 2e, 2f-g, 3b, 3d, 3f, 3h, 3j, 3l, 4b-d, 4h, 5a-b, 6b-d, 7a-b, and 7d-e and Supplementary Figs 1a, 1c-d, 2a-b, 3b-c, and 4a-b are provided as a Source Data file.

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

## Acknowledgements
We thank Akira Nagatani for providing the *PBG* line and phytochrome antibodies, Pablo Cerdán for the *phyABCDE* mutant, Clark Lagarias for the *YHB* line, Peter Quail for the *pifq* mutant, and Sabine Heinhorst for anti-SiR antibodies. We thank Yongjian Qiu, Keunhwa Kim, Joseph Hahm, Ruth Jean Ae Kim, and Jia Wei for valuable suggestions regarding the paper. This work was supported by the National Institute of General Medical Sciences grant R01GM087388 and the National Science Foundation grant IOS-1051602 to M.C.

## Author contributions
C.Y., E.K.P. and M.C. conceived the original research plan; M.C supervised the experiments; C.Y., E.K.P., H.W. and M.C. performed the experiments; C.Y., E.K.P., H.W., G.M.B. and M.C. analyzed the data; J.C. and D.W. carried out the SHOREmap experiments; C.Y., E.K.P., D.W., and M.C. wrote the article with contributions from all the authors.

## Additional information

**Competing interests:** The authors declare no competing interests.

