## [Peer Review File · Nature Communications]

Reviewers' comments:

Reviewer #1 (Remarks to the Author):

Yul Yoo et al., identified Regulator of Chloroplast Biogenesis (RCB) as a dual-targeted nuclear/plastidial phytochrome signaling component required for PEP assembly using a forward genetic screen for the combination of tall and albino seedling phenotypes. RCB controls photosynthetic gene expression in the plastids, primarily from the nucleus, by interacting with phytochromes and promoting their localization to photobodies for the degradation of the transcriptional regulators PIF1 and PIF3. RCB is shown to be dual-targeted, and in the plastids, RCB was shown to facilitate the assembly of PEP into a 1000-kDa complex required for transcription.

The work by Yul Yoo et al. is thorough and the experiments are professionally performed. The text is nicely written and the figures are well presented. However, I have some questions that should be addressed to highlight the novelty of the dual localization and the mechanistic role of RCB especially in the context of the other recently described component RCBL.

Major comments:

1. The principle investigator of this study has previously described HEMERA, HMR (PTAC12), and in addition very recently RCBL (presented in a manuscript that was invited for resubmission) all as dually localized protein involved both in phytochrome signaling and plastid transcription. My major question is how do these newly identified proteins, RCB and RCBL, relate to the action of HMR? As far as I can understand all three proteins seem to have similar functions. How is it possible that there is no redundancy between these components with very similar function? What are their specific roles? This needs to be clarified.

2. The timing of the different localization is critical, what is the timing of RCB localization during the early light response? The data suggest RCB is first imported and processed in the chloroplast to later be translocated to the nucleus. Does this mean RCB is present in the plastids in the dark? Is the translocation from the plastids light triggered? This needs to be determined. Deletion variants without the plastid transit peptide could be used to determine whether RCB could enter the nucleus without coming from the chloroplast.

3. Interestingly, the PEP complex was found to assemble in the dark in the *pifq* mutant. What is the role of RCB on PEP assembly? Does this mean that PEP assembly is solely dependent on the nuclear action of RCB?

4. Is the expression of the other nuclear encoded PEP components (PAPs) impaired in the *rcb* mutant?

Minor comments:

1. More information about the PBG line would be useful for the reader.

2. Line 89 typo, t at the end of the line

3. Line 199. It could be added that the protein does not accumulate in the T-DNA mutant line (Supplementary Fig. 1a)

4. Line 212. PHY triggered is not supported in the section of the results. RCB activates PhAPG and assembly of PEP. The results to support this are in the next section (RCB mediates PHY signal)

5. Line 258, add ref to *fig4d* that includes *hmr*, not only the reference to HRM paper

6. Fig 4f the meaning of the white vs grey bars is not in the legend

Reviewer #2 (Remarks to the Author):

In this work, Yoo and co-authors describe that nuclear phytochrome photoreceptors and PIF transcription factors regulate the assembly of the plastidial RNA polymerase (PEP) complex to initiate chloroplast biogenesis. Through a novel forward genetic screen, they identify and characterize RCB, a novel nucleo-plastidic component that participate in this process by regulating phytochrome photobody formation and PIF degradation.

This manuscript represents an important advance towards understanding the mechanisms of light and phytochrome control of chloroplast biogenesis, establishing a new regulatory connection between nuclear and chloroplast genomes. The question is interesting and novel, and, overall, experiments appear well-performed and data well-analyzed. Although the work is thorough, I have some questions that the authors should address.

A central point to the story is that RCB mainly acts in the nucleus, as *rcb* mutants “fail to degrade” nuclear PIF1 and PIF3. Consistent with this notion, a *pifq* mutation lacking PIF1, PIF3, PIF4 and PIF5, suppresses the long hypocotyl and the albino phenotypes of the *rcb* mutant. Although the data appear convincing, it is intriguing that a chloroplast localized protein do not show any chloroplast-associated function. Authors suggest that knocking out PIFs in the *rcb* mutant can bypass the function of RCB to trigger chloroplast biogenesis, but I believe that this question should be explored in more detail.

(1) First, authors claim that *pifq* mutation fully suppresses the *rcb* mutant phenotype in the *rcb/pifq* mutant, but data are only presented at a just one fluence rate in figure 6a and 6b. I wonder whether this phenotypic suppression is consistently observed over a range of light fluences in a fluence rate response curve.

Additionally, are there any growth or greening alterations in *rcb/pifq* compared to *pifq* later in development that can be attributed to chloroplast-associated/PIF-independent functions of RCB?

(2) Because data indicate that PIFq members act in the dark to repress photomorphogenesis, PEP-assembly, and chloroplast biogenesis, I think it would also be important to explore the genetic interactions between RCB and PIFq in the dark and during the early dark to light transition. For example, can the authors analyze the photomorphogenic phenotype and the PEP complex assembly in the *rcb/pifq* mutant in the dark and upon illumination?

(3) Moreover, in the current form, the gene expression analysis in figure 4 does not seem very informative. Authors cross a list of previously published PIF-target genes, with a list of RCB-regulated genes described in the present study. The main concern is that in previous works the PIFq-directly regulated genes were established in the dark, whereas in this study RCB-regulated genes have been established under continuous light, so that a direct comparison is not possible. In order to test for co-regulation of gene expression by RCB and PIFq, authors can explore the expression of some selected genes in *pifq*, *rcb*, and *rcb/pifq* in the dark and during the dark to light transition.

(4) *Rcb* mutant shows increased PIF1 and PIF3 levels, whereas levels of PIF4 and PIF5 have not been determined in the manuscript. In contrast, genetic analyses are performed with the *pifq* mutant, which lacks PIF4 or PIF5 in addition to PIF1 and PIF3. I think that it cannot be ruled out the formal possibility that the observed *pifq* phenotypes are established by PIF4/5 rather than PIF1/PIF3. Although it is certainly unlikely, authors must at least acknowledge this possibility, or alternatively, include the analysis of *pif1/pif3* rather than *pifq* in the present study.

(5) Authors claim that the *rcb* mutant “fail to degrade PIF1 and PIF3 proteins”. To assess whether there is a complete or a partial failure of degradation in the light, authors must also include PIF1/PIF3 levels in the dark as a reference.

Other minor comments:

(6) Fig 3k and 3l: Authors claim that HMR acts in concert with RCB to promote PIF degradation and activation of PIF target genes. However, phenotypic data of *rcb/hmr* double mutant are not fully consistent with this view, as *hmr* seems to partly suppress the *rcb* phenotype. I wonder whether a fluence rate response curve would be more informative.

(7) Authors claim that *rcb-10/phyB9* double mutant is not taller than *phyB*. Actually, data in Figure 3e indicate that *rcb-10/phyB9* is a bit shorter than *phyB9*. How does this fit with the model?

(8) Fig 3j: Authors claim that *rcb-1* suppresses the *cop*-like phenotype of YHB. I would say that this effect is partial, as it appears to be the result of an incomplete suppression of the phenotype.

(9) Typo at the end of line 89 (a "t" appearing at the end of the line)

(10) Fig 1a legend: YHB lines are not shown in figure 1a/1b.

Response to Reviewers

Reviewer #1

Yul Yoo et al., identified Regulator of Chloroplast Biogenesis (RCB) as a dual-targeted nuclear/plastidial phytochrome signaling component required for PEP assembly using a forward genetic screen for the combination of tall and albino seedling phenotypes. RCB controls photosynthetic gene expression in the plastids, primarily from the nucleus, by interacting with phytochromes and promoting their localization to photobodies for the degradation of the transcriptional regulators PIF1 and PIF3. RCB is shown to be dual-targeted, and in the plastids, RCB was shown to facilitate the assembly of PEP into a 1000-kDa complex required for transcription.

The work by Yul Yoo et al. is thorough and the experiments are professionally performed. The text is nicely written and the figures are well presented. However, I have some questions that should be addressed to highlight the novelty of the dual localization and the mechanistic role of RCB especially in the context of the other recently described component RCBL.

Major comments:

1. The principle investigator of this study has previously described HEMERA, HMR (PTAC12), and in addition very recently RCBL (presented in a manuscript that was invited for resubmission) all as dually localized protein involved both in phytochrome signaling and plastid transcription. My major question is how do these newly identified proteins, RCB and RCBL, relate to the action of HMR? As far as I can understand all three proteins seem to have similar functions. How is it possible that there is no redundancy between these components with very similar function? What are their specific roles? This needs to be clarified.

Response: The main difference between HMR and RCB/RCBL is that HMR (also called pTAC12) is one of the twelve plant-specific PEP accessory proteins. In contrast, although RCB and RCBL could be loosely associated with the nucleoid, they are not part of the closely-associated PEP complex (Steiner et al. 2011 *Plant Physiol* 157:1043-55). Consequently, HMR plays dual essential roles in PIF1/3 degradation in the nucleus as well as PEP assembly in the plastid, whereas RCB acts primarily in the nucleus in promoting PIF1/3 degradation. In the RCBL manuscript, we show that although RCBL is not one of the PEP-associated proteins, it also plays an essential role in PEP assembly in plastids. We have added the differences between HMR and RCB in the second paragraph of the Discussion.

2. The timing of the different localization is critical, what is the timing of RCB localization during the early light response? The data suggest RCB is first imported and processed in the chloroplast to later be translocated to the nucleus. Does this mean RCB is present in the plastids in the dark? Is the translocation from the plastids light triggered? This needs to be determined. Deletion variants without the plastid transit peptide could be used to determine whether RCB could enter the nucleus without coming from the chloroplast.

Response: One of the surprises from our genetic studies is that phytochrome signaling and the PEP are connected by dual-targeted nuclear/plastidial proteins such as HMR and RCB. It is also unexpected that the nuclear and plastidial fractions of HMR and RCB have the same molecular mass, which suggests that these proteins localize to the plastids first before relocating to the nucleus. The mechanism of such a plastid-to-nucleus protein translocation pathway is almost completely unknown. We agree with this reviewer that the regulation of the nuclear/plastidial partitioning of RCB would be very important to further understanding its role in the nucleus-to-plastid signaling. However, these questions require careful assessments. We recently published a study on HMR's dual-localization mechanism (Nevarez et al. 2017 *Plant Physiol* 173:1953-66). As you can see, this study include biochemical and genetic experiments that will take at least two or more years. Therefore, we feel that these studies should be included in a subsequent study.

3. Interestingly, the PEP complex was found to assemble in the dark in the *pifq* mutant. What is the role of RCB on PEP assembly? Does this mean that PEP assembly is solely dependent on the nuclear action of RCB?

Response: The unexpected results that the *rcb-10/pifq* mutant rescued the defects of *rcb-10* in PEP assembly, *PhAPG* expression, and chloroplast biogenesis (Fig. 6) provide genetic evidence supporting the conclusion that RCB regulates plastidial gene expression primarily from the nucleus. However, we can not exclude a direct role of RCB in regulating the PEP in plastids, although its role in plastids is likely non-essential for chloroplast biogenesis.

4. Is the expression of the other nuclear encoded PEP components (PAPs) impaired in the *rcb* mutant?

Response: We thank this reviewer for the suggestion as one possibility is that RCB controls PEP assembly by regulating the expression of the nuclear-encoded PEP-associated proteins (PAPs). We therefore performed qRT-PCR experiments to test whether the expression of any of the *PAPs* was reduced in *rcb-10* compared with Col-0 and rescued to the wild-type level in *rcb-10/pifq*. We have included the results of these experiments in Supplementary Fig. 4. The expression of most *PAPs* was not significantly altered between *rcb-10* and Col-0 and between *rcb-10* and *rcb-10/pifq*.

Minor comments:

1. More information about the *PBG* line would be useful for the reader.

Response: We added the description of the *PBG* line in the Results: "The screen was conducted using *PBG* (PHYB-GFP), a transgenic line in the null *phyB-5* background complemented with functional PHYB-GFP¹⁷. This design allowed us to easily assess whether the diagnostic signaling event of photobody formation is impaired in the mutants."

2. Line 89 typo, *t* at the end of the line

Response: Corrected.

3. Line 199. It could be added that the protein does not accumulate in the T-DNA mutant line (Supplementary Fig. 1a)

Response: We have added the following line describing the supplementary data: “The gene product of At4g28590 did not accumulate in *rcb-10* (Supplementary Fig. 1a), indicating that it is a null allele.”

4. Line 212. *PHY* triggered is not supported in the section of the results. RCB activates *PhAPG* and assembly of *PEP*. The results to support this are in the next section (RCB mediates *PHY* signal)

Response: We have revised the conclusion to “These results indicate that RCB activates *PhAPG* expression by promoting *PEP* assembly.”

5. Line 258, add ref to *fig4d* that includes *hmr*, not only the reference to *HMR* paper

Response: Added.

6. Fig 4f the meaning of the white vs grey bars is not in the legend

Response: We have added the meaning of the white and grey bars in the legend.

Reviewer #2

In this work, Yoo and co-authors describe that nuclear phytochrome photoreceptors and PIF transcription factors regulate the assembly of the plastidial RNA polymerase (PEP) complex to initiate chloroplast biogenesis. Through a novel forward genetic screen, they identify and characterize RCB, a novel nucleo-plastidic component that participate in this process by regulating phytochrome photobody formation and PIF degradation.

This manuscript represents an important advance towards understanding the mechanisms of light and phytochrome control of chloroplast biogenesis, establishing a new regulatory connection between nuclear and chloroplast genomes. The question is interesting and novel, and, overall, experiments appear well-performed and data well-analyzed. Although the work is thorough, I have some questions that the authors should address.

*A central point to the story is that RCB mainly acts in the nucleus, as *rcb* mutants “fail to degrade” nuclear PIF1 and PIF3. Consistent with this notion, a *pifq* mutation lacking PIF1, PIF3, PIF4 and PIF5,*

suppresses the long hypocotyl and the albino phenotypes of the *rcb* mutant. Although the data appear convincing, it is intriguing that a chloroplast localized protein do not show any chloroplast-associated function. Authors suggest that knocking out PIFs in the *rcb* mutant can bypass the function of RCB to trigger chloroplast biogenesis, but I believe that this question should be explored in more detail.

(1) First, authors claim that *pifq* mutation fully suppresses the *rcb* mutant phenotype in the *rcb/pifq* mutant, but data are only presented at a just one fluence rate in figure 6a and 6b. I wonder whether this phenotypic suppression is consistently observed over a range of light fluences in a fluence rate response curve.

Additionally, are there any growth or greening alterations in *rcb/pifq* compared to *pifq* later in development that can be attributed to chloroplast-associated/PIF-independent functions of RCB?

Response: We agree with the reviewer that the surprising and interesting feature of the *rcb* mutant is that knocking-out four nuclear transcription factors (PIF1, PIF3, PIF4, and PIF5) is able to rescue not only the long hypocotyl phenotype (expected) but also the albino phenotype (unexpected). As the reviewer said, these results provide “convincing” genetic evidence supporting the model that RCB regulates *PhAPG* expression primarily from the nucleus by promoting PIF degradation. Because the *pifq* mutant can stimulate PEP assembly and *PhAPG* expression even in the dark, it is expected that *PhAPG* is activated in *rcb-10/pifq* in any light conditions. If RCB plays a direct role in PEP assembly in plastids, the *rcb-10/pifq* mutant should still be albino like *hmr-5/pifq* (Qiu et al. 2015 *Plant Cell* 27:1409-27).

What we claimed in the Results was “Surprisingly, knocking out *PIF1*, *PIF3*, *PIF4*, and *PIF5* in *rcb-10* rescued its long-hypocotyl and albino phenotypes (Fig. 6a, b). Moreover, the *rcb-10/pifq* mutant also rescued *rcb-10*'s defects in PEP assembly and *PhAPG* activation (Fig. 6c, d). These results demonstrate that RCB controls *PhAPG* expression primarily from the nucleus by facilitating PHY-mediated PIF degradation.”

We discussed (in the Discussion) that “the *rcb* mutant represents a unique subgroup of albino mutants whose albinism is not caused by defects in the chloroplast *per se* but rather due to the failure of degrading the nuclear repressors of chloroplast biogenesis (the PIFs) in the light.”

We also indicated in the Discussion that “We do not exclude a direct function for RCB in plastids, where RCB is associated with the PEP complex in the nucleoid⁵¹ and is required for the maintenance of the photosynthetic apparatus⁷⁰. It is possible that the plastidial function of RCB is not essential for the activity of the PEP or that knocking out the *PIFs* in *rcb-10* can bypass these plastidial functions of RCB to trigger chloroplast biogenesis.”

We did not observe any obvious chloroplast-related phenotypic defects in *rcb-10/pifq* in later developmental stages.

(2) *Because data indicate that PIFq members act in the dark to repress photomorphogenesis, PEP-assembly, and chloroplast biogenesis, I think it would also be important to explore the genetic interactions between RCB and PIFq in the dark and during the early dark to light transition. For example, can the authors analyze the photomorphogenic phenotype and the PEP complex assembly in the rcb/pifq mutant in the dark and upon illumination?*

Response: We wanted to make the point that RCB mediates PHY-dependent hypocotyl inhibition and chloroplast biogenesis through the regulation of PIFs in the light. The *rcb* mutant does not have a phenotype in the dark (Supplementary Fig. 2A). Similar to HMR (Qiu et al. 2015 *Plant Cell* 27:1409-27), both RCB and HMR are involved in PHY signaling in the light. The *pifq* mutant can induce PEP assembly (and photomorphogenic responses) even in the dark (Fig. 6), so it would not be informative to look at these mutants during the dark-to-light transition. Therefore, the suggested experiments would not add additional critical information to the genetic evidence we have already provided in Fig. 3 demonstrating that RCB is required for PHY signaling and in Fig. 6 that both the tall and albino phenotypes of *rcb-10* are PIF-dependent.

(3) *Moreover, in the current form, the gene expression analysis in figure 4 does not seem very informative. Authors cross a list of previously published PIF-target genes, with a list of RCB-regulated genes described in the present study. The main concern is that in previous works the PIFq-directly regulated genes were established in the dark, whereas in this study RCB-regulated genes have been established under continuous light, so that a direct comparison is not possible. In order to test for co-regulation of gene expression by RCB and PIFq, authors can explore the expression of some selected genes in pifq, rcb, and rcb/pifq in the dark and during the dark to light transition.*

Response: We thank the reviewer comments. We have previously shown that HMR controls both PIF3 degradation and the expression of a distinct set of PIF-target genes. We proposed the model that in the light HMR's transactivation domain couples the activation of these PIF target genes with PIF3 degradation (Qiu et al. 2015 *Plant Cell* 27:1409-27). Because RCB is also required for PIF3 degradation, we tested whether RCB is involved in the regulation of HMR-regulated, PIF3-dependent genes. To make the points more clear, we have moved the data from the original Supplementary Fig. 4 to Fig. 4e, g. Here we show that 871 or 88% of RCB-dependent genes were also HMR-dependent (Fig. 4e). Most of the PIF-induced genes are downregulated in *rcb-10* (Fig. 4f), and HMR and RCB coregulate the same sets of genes (Fig. 4g). In particular, we show qRT-PCR results of selected PIF-induced, RCB/HMR-induced genes (Fig. 4h). We have previously shown that the expression of these marker genes is PIF-dependent in the light (downregulated in *pifq*), therefore, these are PIF-regulated genes under our experimental condition (Qiu et al. 2015 *Plant Cell* 27:1409-27).

(4) *Rcb mutant shows increased PIF1 and PIF3 levels, whereas levels of PIF4 and PIF5 have not been determined in the manuscript. In contrast, genetic analyses are performed with the pifq mutant, which lacks PIF4 or PIF5 in addition to PIF1 and PIF3. I think that it cannot be ruled out the formal possibility that the observed pifq phenotypes are established by PIF4/5 rather than PIF1/PIF3. Although it*

is certainly unlikely, authors must at least acknowledge this possibility, or alternatively, include the analysis of pif1/pif3 rather than pifq in the present study.

Response: We have revised the text and added the following in the Discussion: “Here we show that the *rcb* mutants, which accumulated PIF1 and PIF3 in the light (Fig. 4d), failed to stimulate PEP assembly and *PhAPG* expression in the light (Fig. 2g, f). Both PEP assembly and *PhAPG* expression were de-repressed in the *pifq* mutant in the dark (Fig. 5) as well as in *rcb-10/pifq* in the light (Fig. 6c, d), indicating that PIFs are nuclear repressors of *PhAPG* expression. It is worth noting that it remains unclear whether all PIFs are required for repressing *PhAPG* expression, which requires further genetic analysis of the combinations of triple *pif* mutants in the future. Nonetheless, our results demonstrate that PHY- and RCB-mediated PIF degradation is a central nuclear event that triggers *PhAPG* expression in plastids.”

(5) *Authors claim that the rcb mutant “fail to degrade PIF1 and PIF3 proteins”. To assess whether there is a complete or a partial failure of degradation in the light, authors must also include PIF1/PIF3 levels in the dark as a reference.*

Response: We have revised the text and change the sentence in the Results to: “Interestingly, similar to *hmr-5*, PIF1 and PIF3 accumulated or failed to be completely degraded in *rcb-1/PBG* and *rcb-10* in the light (Fig. 4d)”. In this experiment, we included both the parental types as positive controls (lines that PIF1 and PIF3 cannot accumulation in the light) and *hmr-5* as a negative control (a line that PIF1 and PIF3 accumulate in the light). The above statement was made based on the comparisons to the positive and negative controls.

Other minor comments:

(6) *Fig 3k and 3l: Authors claim that HMR acts in concert with RCB to promote PIF degradation and activation of PIF target genes. However, phenotypic data of rcb/hmr double mutant are not fully consistent with this view, as hmr seems to partly suppress the rcb phenotype. I wonder whether a fluence rate response curve would be more informative.*

Response: If RCB and HMR work in parallel pathways, we would expect to see an additive effect in the *hmr-5/rcb-10* double mutant or the double mutant should have been taller than the single mutants. We did not observe such an additive effect, indicating that RCB and HMR work in the same pathway. We revised the text to the following: “To test whether RCB works in the same signaling pathway as HMR, we generated a *rcb-10/hmr-5* double mutant. The *rcb-10/hmr-5* double mutant was not taller than either *rcb-10* or *hmr-5* (Fig. 3k, l), suggesting that RCB and HMR function in the same PHY-dependent pathway. Interestingly, *rcb-10/hmr-5* had the same hypocotyl length as *hmr-5* and was slightly shorter than *rcb-10* (Fig. 3k, l), which might suggest that for hypocotyl regulation, *hmr-5* is epistatic to *rcb-10*.”

(7) *Authors claim that rcb-10/phyB9 double mutant is not taller than phyB. Actually, data in Figure 3e indicate that rcb-10/phyB9 is a bit shorter than phyB9. How does this fits with the model?*

Response: Similar to the above question, here we wanted to test whether RCB acts in PHYB signaling. If RCB and PHYB worked in a parallel pathway, we would expect to see an additive effect in *rcb-10/phyB-9*, meaning that *rcb-10/phyB-9* should have been taller than *rcb-10* and *phyB-9*. The fact that *rcb-10/phyB-9* was not taller than the single mutants indicates that RCB works in PHYB signaling. We still do not have an explanation why the *rcb-10/phyB-9* was slightly shorter than *phyB-9*. But, the main conclusion is further supported by the genetic evidence from the *rcb-1/YHB* and *rcb-10/hmr-5* mutants.

(8) *Fig 3j: Authors claim that rcb-1 suppresses the cop-like phenotype of YHB. I would say that this effect is partial, as it appears to be the result of an incomplete suppression of the phenotype.*

Response: We have changed the sentence to “The constitutive photomorphogenetic phenotype of *YHB* in the dark was partially suppressed the *rcb-1/YHB* double mutant (Fig. 3i, j), ...”

(9) *Typo at the end of line 89 (a “t” appearing at the end of the line)*

Response: Corrected.

(10) *Fig 1a legend: YHB lines are not shown in figure 1a/1b.*

Response: Corrected.

REVIEWERS' COMMENTS:

Reviewer #1 (Remarks to the Author):

In the revised version of Yang et al., the authors have addressed some of the criticisms I raised on the last version of the manuscript. However, my major concerns with the work were left unanswered.

1. The principle investigator of this study has previously described HEMERA, HMR (PTAC12), and in addition very recently RCBL (presented in a manuscript that was invited for resubmission) all as dually localized protein involved both in phytochrome signaling and plastid transcription. My major question is how do these newly identified proteins, RCB and RCBL, relate to the action of HMR? As far as I can understand all three proteins seem to have similar functions. How is it possible that there is no redundancy between these components with very similar function? What are their specific roles? This needs to be clarified. Response: The main difference between HMR and RCB/RCBL is that HMR also called pTAC12 is one of the twelve plant specific PEP accessory protein. In contrast although RCB and RCBL could be loosely associated with the nucleoid they are not part of the closely associated PEP complex (Steiner et al 2011 Plant Physiol). Consequently, HMR plays dual essential roles in PIF1/3 degradation in the nucleus as well as PEP assembly in the plastid whereas RCB acts primarily in the nucleus in promoting PIF1/3 degradation. In the RCBL manuscript we show that although RCBL is not one of the PEP associated proteins it also plays an essential role in PEP assembly in plastids. We have added the differences between HMR and RCB in the second paragraph of the Discussion.

- Some experimental data would enlighten if these proteins interact, are genetically linked etc.

2. The timing of the different localization is critical, what is the timing of RCB localization during the early light response? The data suggest RCB is first imported and processed in the chloroplast to later be translocated to the nucleus. Does this mean RCB is present in the plastids in the dark? Is the translocation from the plastids light triggered? This needs to be determined. Deletion variants without the plastid transit peptide could be used to determine whether RCB could enter the nucleus without coming from the chloroplast. Response: One of the surprises from our genetic studies is that phytochrome signaling and the PEP are connected by dual-targeted nuclear/plastidial proteins such as HMR and RCB. It is also unexpected that the nuclear and plastidial fractions of HMR and RCB have the same molecular mass, which suggests that these proteins localize to the plastids first before relocating to the nucleus. The mechanism of such a plastid-to-nucleus protein translocation pathway is almost completely unknown. We agree with this reviewer that the regulation of the nuclear/plastidial partitioning of RCB would be very important to further understanding its role in the nucleus-to-plastid signaling. However, these questions require careful assessments. We recently published a study on HMR's dual-localization mechanism (Nevarez et al. 2017 Plant Physiol 173:1953-66). As you can see, this study include biochemical and genetic experiments that will take at least two or more years. Therefore, we feel that these studies should be included in a subsequent study.

- It is critical to address this point to provide further understanding of the action of these proteins. Questions such as where is the protein in the dark? Is it moving to the nucleus in response to light? What about in mature plants etc.? Those questions could be addressed using the tools already available in the laboratory as shown in Figure 3.

3. Interestingly, the PEP complex was found to assemble in the dark in the pifq mutant. What is the role of RCB on PEP assembly? Does this mean that PEP assembly is solely dependent on the RCB nuclear action? Response: The unexpected results that the rcb-10/pifq mutant rescued the defects of rcb-10 in PEP assembly, PhAPG expression, and chloroplast biogenesis (Fig. 6) provide

genetic evidence supporting the conclusion that RCB regulates plastidial gene expression primarily from the nucleus. However, we cannot exclude a direct role of RCB in regulating the PEP in plastids, although its role in plastids is likely non-essential for chloroplast biogenesis.

- Please extend the discussion around this in the manuscript.

4. Is the expression of the other nuclear encoded PEP components (PAPs) impaired in the *rcb* mutant? Response: We thank this reviewer for the suggestion as one possibility is that RCB controls PEP assembly by regulating the expression of the nuclear-encoded PEP-associated proteins (PAPs). We therefore performed qRT-PCR experiments to test whether the expression of any of the PAPs was reduced in *rcb-10* compared with Col-0 and rescued to the wild-type level in *rcb-10/pifq*. We have included the results of these experiments in Supplementary Fig. 4. The expression of most PAPs was not significantly altered between *rcb-10* and Col-0 and between *rcb-10* and *rcb-10/pifq*.

- Thank you. It was a bit unexpected but interesting results.

I am satisfied with the corrections to my minor points.

Reviewer #2 (Remarks to the Author):

Minor additional comments:

I still think that it would be informative to study the PEP assembly in the dark and during the very early dark to light transition in *pifq* compared to *rcb-10/pifq* mutants, to test whether the PEP assembly observed in the *pifq* mutant in the dark requires RCB (similar to the experiments performed in figure 1h for *phyAB* mutants). Nevertheless, I do not think that this aspect affects the main conclusions supported by the data, and I understand that uncovering a function for the plastidial form of RCB may require in deep analysis that is out of the scope of the current manuscript. In the current form, the data already represent an important advance towards understanding phytochrome regulation of chloroplast biogenesis.

Reviewer #1

In the revised version of Yang et al., the authors have addressed some of the criticisms I raised on the last version of the manuscript. However, my major concerns with the work were left unanswered.

1. *The principle investigator of this study has previously described HEMERA, HMR (PTAC12), and in addition very recently RCBL (presented in a manuscript that was invited for resubmission) all as dually localized protein involved both in phytochrome signaling and plastid transcription. My major question is how do these newly identified proteins, RCB and RCBL, relate to the action of HMR? As far as I can understand all three proteins seem to have similar functions. How is it possible that there is no redundancy between these components with very similar function? What are their specific roles? This needs to be clarified.*

Response: The main difference between HMR and RCB/RCBL is that HMR also called pTAC12 is one of the twelve plant specific PEP accessory protein. In contrast although RCB and RCBL could be loosely associated with the nucleoid they are not part of the closely associated PEP complex (Steiner et al 2011 Plant Physiol). Consequently, HMR plays dual essential roles in PIF1/3 degradation in the nucleus as well as PEP assembly in the plastid whereas RCB acts primarily in the nucleus in promoting PIF1/3 degradation In the RCBL manuscript we show that although RCBL is not one of the PEP associated proteins it also plays an essential role in PEP assembly in plastids. We have added the differences between HMR and RCB in the second paragraph of the Discussion.

- *Some experimental data would enlighten if these proteins interact, are genetically linked etc.*

Response: Although we still do not know whether RCB interacts directly with HMR, we have shown genetic evidence that RCB and HMR work in the same genetic pathway of phytochrome signaling (Fig. 3k, l). In addition, we showed that both RCB and HMR participate in PHYB-mediated PIF1 and PIF3 degradation (Fig. 4d) as well as the regulation of PIF-target genes (Fig. 4e-h). We plan to investigate the biochemical function of RCB in phytochrome signaling in future investigations.

2. *The timing of the different localization is critical, what is the timing of RCB localization during the early light response? The data suggest RCB is first imported and processed in the chloroplast to later be translocated to the nucleus. Does this mean RCB is present in the plastids in the dark? Is the translocation from the plastids light triggered? This needs to be determined. Deletion variants without the plastid transit peptide could be used to determine whether RCB could enter the nucleus without coming from the chloroplast.*

Response: One of the surprises from our genetic studies is that phytochrome signaling and the PEP are connected by dual-targeted nuclear/plastidial proteins such as HMR and RCB. It is also unexpected that the nuclear and plastidial fractions of HMR and RCB have the same molecular mass, which suggests that these proteins localize to the plastids first before relocating to the nucleus. The mechanism of such a plastid-to-nucleus protein translocation pathway is almost completely unknown. We agree with this reviewer that the regulation of the nuclear/plastidial partitioning of RCB would be very important to further understanding its role in the nucleus-to-plastid signaling. However, these questions require careful assessments. We recently

published a study on HMR's dual-localization mechanism (Nevarez et al. 2017 *Plant Physiol* 173:1953-66). As you can see, this study include biochemical and genetic experiments that will take at least two or more years. Therefore, we feel that these studies should be included in a subsequent study.

- *It is critical to address this point to provide further understanding of the action of these proteins. Questions such as where is the protein in the dark? Is it moving to the nucleus in response to light? What about in mature plants etc.? Those questions could be addressed using the tools already available in the laboratory as shown in Figure 3.*

Response: We agree with this reviewer that it is important to look into the regulation of RCB's dual localization. However, we also think that these questions should be carefully examined in future investigations because the scope of this study is to (1) define a nucleus-to-plastid signaling and to (2) demonstrate the role of RCB in the nucleus-to-plastid signaling. In contrast to the published results, this study demonstrates that RCB is dual-targeted to plastids and the nucleus (Fig. 4a-c). More importantly, we provided convincing genetic evidence supporting that RCB controls the PEP by regulating PIF degradation in the nucleus (Fig. 5 and 6). It is our opinion that the regulatory mechanism of RCB localization is not required for drawing the conclusions of the current study, and thus, can be addressed in future investigations.

3. *Interestingly, the PEP complex was found to assemble in the dark in the pifq mutant. What is the role of RCB on PEP assembly? Does this mean that PEP assembly is solely dependent on the RCB nuclear action?*

Response: *The unexpected results that the rcb-10/pifq mutant rescued the defects of rcb-10 in PEP assembly, PhAPG expression, and chloroplast biogenesis (Fig. 6) provide genetic evidence supporting the conclusion that RCB regulates plastidial gene expression primarily from the nucleus. However, we cannot exclude a direct role of RCB in regulating the PEP in plastids, although its role in plastids is likely non-essential for chloroplast biogenesis.*

- *Please extend the discussion around this in the manuscript.*

Response: We have emphasized this point in the Discussion.

4. *Is the expression of the other nuclear encoded PEP components (PAPs) impaired in the rcb mutant?*

Response: *We thank this reviewer for the suggestion as one possibility is that RCB controls PEP assembly by regulating the expression of the nuclear-encoded PEP-associated proteins (PAPs). We therefore performed qRT-PCR experiments to test whether the expression of any of the PAPs was reduced in rcb-10 compared with Col-0 and rescued to the wild-type level in rcb-10/pifq. We have included the results of these experiments in Supplementary Fig. 4. The expression of most PAPs was not significantly altered between rcb-10 and Col-0 and between rcb-10 and rcb-10/pifq.*

- *Thank you. It was a bit unexpected but interesting results.*

I am satisfied with the corrections to my minor points.

Reviewer #2

Minor additional comments:

*I still think that it would be informative to study the PEP assembly in the dark and during the very early dark to light transition in *pifq* compared to *rcb-10/pifq* mutants, to test whether the PEP assembly observed in the *pifq* mutant in the dark requires RCB (similar to the experiments performed in figure 1h for *phyAB* mutants). Nevertheless, I do not think that this aspect affects the main conclusions supported by the data, and I understand that uncovering a function for the plastidial form of RCB may require in deep analysis that is out of the scope of the current manuscript. In the current form, the data already represent an important advance towards understanding phytochrome regulation of chloroplast biogenesis.*

Response: We appreciate these comments. We plan to look into PEP assembly in dark-grown *pifq* and *rcb-10/pifq* lines in future investigations.